

# Resource and physiological constraints on global crop production enhancements from atmospheric particulate matter and nitrogen deposition

Luke D. Schiferl[1], Colette L. Heald[1,2], and David Kelly[3]

[1]Department of Civil and Environmental Engineering, Massachusetts Institute of Technology, Cambridge, Massachusetts, USA
[2]Department of Earth, Atmospheric and Planetary Sciences, Massachusetts Institute of Technology, Cambridge, Massachusetts, USA
[3]University of Chicago Computation Institute, Chicago, IL, USA

*Correspondence to*: Luke D. Schiferl (schiferl@mit.edu)

**Abstract.** Changing atmospheric composition, induced primarily by industrialization and climate change, can impact plant health and may have implications for global food security. Atmospheric particulate matter (PM) can enhance crop production through the redistribution of light from sunlight to shaded leaves. Nitrogen transported through the atmosphere can also increase crop production when deposited onto cropland by reducing nutrient limitations in these areas. We employ a crop model (pDSSAT), coupled to input from an atmospheric chemistry model (GEOS-Chem), to predict the impact of PM and nitrogen deposition on crop production. In particular, the crop model considers the resource and physiological restrictions to enhancements in growth from these atmospheric inputs. We find that the global enhancement in crop production due to PM in 2010 under the most realistic scenario is 2.3 %, 11.0 %, and 3.4 % for maize, wheat, and rice, respectively. These crop enhancements are smaller than those previously found when resource restrictions were not accounted for. Using the same model setup, we assess the effect of nitrogen deposition on crops and find modest increases (~2 % in global production for all three crops). This study highlights the need for better observations of the impacts of PM on crop growth and the cycling of nitrogen throughout the plant-soil system to reduce uncertainty in these interactions.

## 1 Introduction

Population growth is intensifying stress on global food production. Simultaneously, anthropogenic activities are changing many aspects of the earth system. This reinforces the need to better understand how crop production may be affected by changes to the water, air, light, and soil required for efficient growth. For example, Challinor et al. (2014) suggest a global decline in crop yield of more than 10 % by 2050 is likely due to climate change. This is uncertain, however, and the projected sign and magnitude varies by crop and region due to localized changes in factors such as temperature and precipitation combined with global carbon dioxide ($CO_2$) enhancement (IPCC, 2014). Several studies have explored the impacts of climate




and air quality on crop production, but this has generally been done without considering physiological limitations and other environmental stresses (e.g., water and nutrients) (Greenwald et al., 2006; Shindell et al., 2011; Tai et al., 2014).

Atmospheric particulate matter (PM), emitted through combustion and natural processes and formed through chemical

oxidation in the atmosphere, is the leading cause of air quality issues globally and is responsible for over 4 million premature deaths per year (Cohen et al., 2017). PM also impacts crop production by modifying shortwave radiation reaching the surface. Through the scattering of light, PM decreases the total shortwave (SW) radiation at the surface, which is made up of direct and diffuse light (SW = direct + diffuse). PM also increases the diffuse fraction (DF) of this SW radiation (DF = $\frac{\text{diffuse}}{\text{SW}}$ ). Increased DF more evenly distributes light throughout the canopy of a plant, redirecting light away from (at times over-saturated) leaves

in direct sunlight and onto shaded leaves. In this way, plants can more efficiently make use of incoming solar radiation (Kanniah et al., 2012). In the case of crops, PM can increase growth and production when the increase in efficiency outweighs the loss of SW radiation. Greenwald et al. (2006) use relationships between DF and a crop's radiative use efficiency (RUE), a measure of how effective a plant converts light into carbon, from Sinclair et al. (1992) along with varying meteorology and a crop model to estimate the impact of PM on crop yield. Assuming no restrictions on growth due to stresses at several sites,

they find a large variation in impacts based on the DF-to-ΔRUE relationship chosen. Under the maximum relationship, maize increases by 0–10 %, wheat increases by 0–5 % and rice increases by 0–40 % under varying cloud conditions. Using this approach, Schiferl and Heald (2017) estimate a global positive impact of PM of 12 %, 16 %, and 9 % on maize, wheat, and rice, respectively, for the year 2010.

Industrial agriculture, driven by the need to produce food for a growing human population, has modified the global nitrogen (N) cycle. By artificially fixing inert nitrogen gas into reactive forms, humans have increased the fluxes of nitrogen throughout the environment, including in the atmosphere, on land, and in the water (Galloway and Cowling, 2002). Nitrogen species in the atmosphere, both reduced and oxidized, return to the surface through deposition processes after being transported away from source regions. Anthropogenic influences on these fluxes change the nitrogen balance in land and water ecosystems. In

natural systems, this can cause acidification and eutrophication, which negatively impacts the biosphere (Beem et al., 2010; Erisman et al., 2007). The deposited nitrogen can also impact crop production, by providing additional fertilization, increasing yields in areas which are nitrogen limited (Goulding et al., 1998). While Ladha et al. (2016) estimate that 13 % of N contained in global maize, wheat, and rice comes from deposited nitrogen, to date, there has been no global study of the change of yield associated with this effect.


PM and nitrogen deposition are also connected: The release of excess nitrogen from fertilizer application and livestock production in the form of ammonia ($NH_3$) contributes to PM formation in the atmosphere under acidic conditions (Seinfeld and Pandis, 2006). Nitric acid ($HNO_3$), an oxidized form of nitrogen oxides ($NO_x$) emissions from mobile and industrial



sources, contributes both to the nitrogen burden and these acid conditions. Nitrogen can also be incorporated in PM as organic nitrates when biogenic volatile organic compounds (BVOCs) react with $NO_x$ (Mao et al., 2013).

Schiferl and Heald (2017) quantify the impact that air quality (ozone and PM) has on current and future global crop production. This analysis, while consistent with the approach generally applied to estimate air quality impacts on crops in previous studies (e.g., Shindell et al., 2011; Tai et al., 2014; Van Dingenen et al., 2009), fails to account for the set of physical and biological restrictions placed on crop growth and production. In particular, crop production enhancement due to the diffuse effect of PM is considered to be unlimited. However water and nitrogen stresses and physiological caps placed on crop production may dampen these responses. In this study, we employ a crop production model to simulate the enhancements in crop production associated with PM and nitrogen deposition and explore these constraints.

## 2 GEOS-Chem Atmospheric Chemistry Model

The GEOS-Chem model (www.geos-chem.org) simulates the global concentration of gases and particles in three dimensions. Simulated PM concentrations are read into the Rapid Radiative Transfer Model for GCMs (RRTMG) to estimate the impact of PM on radiation throughout the atmosphere (Heald et al., 2014). Together these models are referred to as GC-RT. The model version and setup used here is the same as for the standard 2010 emissions scenario described by Schiferl and Heald (2017). In brief: GC-RT is run at $2° \times 2.5°$ horizontal resolution using GEOS-5 meteorology for the years 2009 and 2010 from the NASA Global Modeling and Assimilation Office (GMAO). Inorganic aerosol thermodynamics are coupled to an ozone–VOC–$NO_x$–oxidant chemical mechanism, where ISORROPIA II (Fountoukis and Nenes, 2007) handles the gas-particle phase partitioning of ammonium nitrate. GC-RT simulates wet and dry deposition of both aerosols and gases (Amos et al., 2012; Liu et al., 2001; Wang et al., 1998; Zhang et al., 2001). Major global anthropogenic gas emissions come from the Emission Database for Global Atmospheric Research (EDGAR) v4.2 ($NO_x$, carbon monoxide (CO), sulfur dioxide ($SO_2$)) (edgar.jrc.ec.europa.eu), the Reanalysis of the TROpospheric chemical composition (RETRO) inventory (non-methane VOCs) (Hu et al., 2015), and the Global Emission Inventory Activity (GEIA) inventory ($NH_3$). These are overlaid by regional inventories where available (see Schiferl and Heald (2017) for details). Additional $NO_x$ emissions are from lightning and soil, described by Murray et al. (2012) and Hudman et al. (2012), respectively. Directly emitted aerosol sources include anthropogenic black carbon (BC) and organic carbon (OC) (Bond et al., 2007; Leibensperger et al., 2012), dust (Fairlie et al., 2007), and sea salt (Jaeglé et al., 2011).

In this study, we use hourly output of surface SW radiation and the diffuse and direct portions of this SW radiation from GC-RT both with and without PM under all-sky (real time variation in cloudiness) conditions. These are used to calculate the DF of the SW radiation. PM refers to the sum of all simulated aerosol species: sulfate ($SO_4^{2-}$), nitrate ($NO_3^-$), ammonium ($NH_4^+$), black carbon, organic carbon, sea salt and dust. We also use daily output of nitrogen deposition flux, including the wet and dry



deposition simulated for all nitrogen species. Nitrogen mass deposited from five species, ammonia, ammonium, nitric acid, nitrate, and nitrogen dioxide ($NO_2$), make up 98 % of the total simulated nitrogen deposition for 2010.

## 3 pDSSAT Crop Model

### 3.1 Model Description

We use v4.6 of the Decision Support System for Agrotechnology Transfer (DSSAT) crop system model (www.dssat.net), along with v2.0 of the parallel System for Integrating Impact Models and Sectors (pSIMS) (www.github.com/RDCEP/psims), together called pDSSAT, to simulate the global production of maize, wheat, and rice. DSSAT provides a unified interface which combines various crop simulation models (Jones et al., 2003). Inherently a point model, DSSAT uses daily meteorological data (minimum temperature, maximum temperature, precipitation, solar radiation, wind speed, and relative

humidity) along with soil and management information at a given location. The model then calculates a crop yield at harvest taking into account soil-plant-atmosphere dynamics throughout the growing season. Plant growth, in our case, is determined by the Crop-Environment Resource Synthesis (CERES) model module for each crop (Jones et al., 1986; Ritchie et al., 1998; Ritchie and Otter, 1985).

pSIMS allows for the globally gridded simulation of crop yield by running DSSAT in parallel at various grid boxes using consistent data and setting input methods (Elliott et al., 2014). In our study, we set pDSSAT to run at 0.5° × 0.5° horizontal resolution. This is only limited by the availability of suitable global input data. pDSSAT uses daily meteorological information from AgMERRA (Ruane et al., 2015), a version of the NASA Modern-Era Retrospective Analysis for Research and Applications (MERRA) product developed for use in the Agricultural Model Intercomparison and Improvement Project

(AgMIP) (Rosenzweig et al., 2013). We note that this meteorological product is closely related to the GEOS-5 product which drives the GC-RT simulations. Soil inputs come from the Global Soil Dataset for Earth System Modeling (GSDE) (Shangguan et al., 2014). Additional required information includes the range of planting dates (Portmann et al., 2010; Sacks et al., 2010), distribution of cultivars (based on local growing degree days (GDD)), and fertilizer application amounts (You et al., 2012) at each grid box. Except for the soil inputs, which are modified in pSIMS v2.0, these data are consistent with those used by the

global gridded crop model (GGCM) intercomparison portion of AgMIP (Rosenzweig et al., 2014).

### 3.2 Integration of GEOS-Chem with pDSSAT

Using the hourly SW, diffuse and direct radiation output from GC-RT, we calculate the daily mean daytime (SW > 0) SW and DF for each GEOS-Chem gridbox (2° × 2.5° horizontal resolution) for all of 2009 and 2010. We group the nitrogen deposition fluxes of individual species into two groups, reduced nitrogen ($NH_x$) and oxidized nitrogen ($NO_y$), and calculate the daily total

flux for each group for the same time period. The daily SW and DF values, along with the daily $NH_x$ and $NO_y$ deposition flux values, are regridded to the pDSSAT resolution and integrated into the input meteorology.



For the PM simulations, the daily SW and DF are used in the pDSSAT crop-specific plant growth modules to modify the potential carbon production. Equation 1 is used for maize and wheat, and Eq. 2 is used for rice:

$$P_{carb} \propto 0.5 \times SW \times RUE_{s,DF} \tag{1}$$

$$P_{carb} \propto (0.5 \times SW)^{0.65} \times RUE_{s,DF} \tag{2}$$

where $P_{carb}$ is the potential carbon production, SW is the daily mean shortwave radiation from GC-RT, and $RUE_s$ is crop-specific radiation use efficiency (Ritchie et al., 1998). For simulations with PM affecting SW and DF, SW modified by PM from GC-RT is used as input for these relationships only and is not used in other functions dependent on solar radiation, such as evaporation (i.e., the GC-RT SW without PM remains applied to these processes). In this study, we apply only the maximum DF-to-$\Delta$RUE relationship discussed in Schiferl and Heald (2017), where max $\Delta$RUE = 100 % at DF = 0.8 (Greenwald et al., 2006). This represents the upper-limit of potential PM impacts on crop production.

For the nitrogen deposition simulations, $NH_x$ and $NO_y$ fluxes are applied daily as fertilizer to the surface layer of the soil as $NH_4^+$ and $NO_3^-$, respectively, due to their similar behaviors in soils (Ladha et al., 2016). We apply these deposition fluxes beginning 30 days prior to the planting date at each location. The timing of this initiation is uncertain, as the fate of deposited nitrogen is not well constrained, and the impacts of nitrogen deposition can be assessed over a single growing season to multi-year time scales (Goulding et al., 1998). We discuss the impact of this assumption in Sect. 4.2.

### 3.3 Base Simulation

We configure pDSSAT to run for 2009 and 2010 with water and nitrogen stress turned off. Our modification for potential carbon production using input from GC-RT is applied to SW only (with SW values from GC-RT without PM). Maize, wheat, and rice are simulated independently. We sample the results for each crop for the growing season ending in 2010. For example, crops planted in northern hemisphere spring and harvested in fall are grown entirely within 2010, while winter crops are planted in fall 2009 and harvested in spring 2010. For a consistent comparison, we determine crop production by multiplying the pDSSAT crop yield by the crop area from the Global Agro-Ecological Zones (GAEZ) assessment for 2000 (www.fao.org/nr/gaez) scaled to 2010 as in Schiferl and Heald (2017), rather than by using the internal pDSSAT harvested area parameter. The results from this simulation, our base simulation, are shown in Fig. 1. Also as in Schiferl and Heald (2017), we focus our figures on the industrialized areas of the northern hemisphere, which rely heavily on maize, wheat, and rice, though all numbers presented are global. Since our base simulation has no restrictions on water and nitrogen (both the nitrogen supply and irrigation are unlimited), the simulated crop production vastly surpasses that from GAEZ. For maize, this is 2062 Tg from pDSSAT compared to only 871 Tg from GAEZ. Simulated wheat production is 2591 Tg, and simulated rice production is 1250 Tg compared to GAEZ values of 667 Tg and 705 Tg, respectively.





We rerun the crop model with water stress only, nitrogen stress only, and both stresses together to test the sensitivity of the base simulation to these resources (Fig. 1). Water stress occurs when the amount of soil water available is below the potential transpiration rate of the plant. For maize, the negative effect of water stress on production is most evident in the United States (US) Plains and northern China and produces a 29 % production reduction globally. The effect of water stress is larger globally

on wheat (40 % reduction), and is largest in the southern US Plains, northern China, and throughout western Asia. Rice production is impacted the least by water stress, with only a 14 % reduction in production when imposing water stress, mostly in northern India. Water stress is dependent on the precipitation prescribed from the meteorology of that growing season, so these results will vary from year to year.

Nitrogen stress occurs when the plant tissue nitrogen concentration is less than the critical nitrogen concentration determined to provide optimal growth. In our base simulation, nitrogen stress follows different patterns compared to water stress for many regions and crops, although the global magnitudes in production reduction are similar. Maize production is affected by nitrogen stress primarily in the US Plains and Midwest. Nitrogen stress for wheat is distributed into all regions, while the effect on rice production is again lowest and largest in southeast (SE) Asia. Nitrogen stress is more similar from year to year in the model

as fertilizer application, which provides nitrogen to the soil, and inherent soil nitrogen content is identical for all simulation years. Small variations do exist as variable temperatures and radiation impact the onset of crop growth stages and use of nitrogen. Total production change due to both water and nitrogen stress does not combine linearly. This illustrates the interconnected system simulated by the crop model. Overall, these environmental and management constraints greatly reduce global crop production from its unstressed potential. They are important to consider when analyzing the impact of PM and

nitrogen deposition on crop production.

## 4 Results

### 4.1 Impact of Particulate Matter on Crop Growth

To simulate the effect of PM on crop production, we run pDSSAT as above with SW and DF input from GC-RT with and without PM. The differences in SW and DF due to PM over the pDSSAT growing season are shown in Fig. 2. PM has a

negative effect on SW everywhere and positive effect on DF. The largest influence of PM is over China for all three crops. The influence is especially noticeable for wheat, where a growing season over the winter corresponds with higher PM concentrations. The difference between the simulations with and without PM is the change in production due to PM, and this is shown in Fig. 3. We perform this procedure first with no stress factors applied. Under no stress, global maize production increases by 1.7 %, wheat increases by 17.0 %, and rice increases by 6.2 %. Wheat production in the India and China+SE Asia

regions is most affected by PM, and the regional proportional change is show in Fig. 4. For wheat and rice, pDSSAT reproduces well the proportional enhancement in crop production due to PM found in Schiferl and Heald (2017) using an offline relativistic



methodology (Fig. 4). This is true globally and within each region. The pDSSAT scenario with no stress is closely related to the offline analysis, which was unrestricted in production enhancement, so this good comparison is expected.

However, the proportional increase in maize production due to PM simulated by the pDSSAT model is much lower than that
from the offline analysis. This can be explained by a physiological restriction within the model which limits the maximum number of kernels per maize plant based on its genetic potential. Within pDSSAT, hybrid cultivars are limited to about 900 kernels per plant, while open-pollinated cultivars are limited to about 550 kernels per plant. In the scenario above with no stress, PM only produces a 1.2 % increase in maize production over the US (Fig. 5). For most locations in this domain, pDSSAT simulates the maximum maize production dictated by the kernel number both with and without PM. When we artificially
increase the limit by 500 kernels per plant, the maize production increases, as expected. Production without PM increases by 25 %, and production with PM increases by 34 %. This results in an 8.4 % increase in maize production due to PM over the US under no stress, which is similar to the approximately 10 % increase found in the offline analysis. This dependence on a kernel limit demonstrates the importance of including physiological limitations to growth as represented in a crop production model when addressing the air quality impacts on crops.

To investigate the more realistic effect of PM on crop production, we impose both water and nitrogen stress on our pDSSAT simulations. The results for this scenario (Figs. 3 and 4) indicate an 11 % increase in global wheat production due to PM and a 3.4 % increase in rice. These proportional enhancements are about one-third lower with stresses for wheat compared to without and about one-half for rice. While similar declines occur on a regional basis, these stresses have a larger impact on
India for wheat, where nearly one-half of additional simulated wheat production is lost. For maize, including stress factors under the standard kernel restriction lowers the total production with and without PM, but allows for a larger proportional change due to PM in most areas (i.e., 1.7 % global production increase without stress, but 2.3 % with stresses) as more areas are producing below the production limit. When additional kernels are allowed with stresses turned on, production due to also PM increases, but to a lesser percentage compared to without stress (not shown).

**4.2 Impact of Nitrogen Deposition on Crop Growth**

We run pDSSAT as above with additional $NH_x$ and $NO_y$ atmospheric deposition fluxes from GEOS-Chem and compare the results to the base simulation to quantify the impact of nitrogen deposition on crop production. No PM effects are considered. The total nitrogen deposition flux for each crop over the base scenario growing season is shown in Fig. 6. There is high nitrogen deposition in India and China for all three crops, but especially wheat in China. The magnitude of nitrogen deposition from
GEOS-Chem is generally lower than that applied as fertilizer. For example, two fertilizer applications for maize span roughly 50–100 kg ha$^{-1}$ each over the US, Europe and China, whereas nitrogen deposition during the growing season rarely exceeds 20 kg ha$^{-1}$ in China and India. However, nitrogen deposition is continuous, while fertilizer application is sporadic and limited temporally. We also plot the fraction of total nitrogen deposition made up of $NH_x$ in Fig. 6. This fraction is slightly higher in



agricultural areas of the US, Europe and China, where reduced species from agriculture mix with oxidized species from industry. In India, the $NH_x$ fraction is very high, as there is little industrial emission to offset the large agricultural emissions.

When accounting for both nitrogen and water stress, crop production increases globally by 1.9 % for maize, 1.8 % for wheat, and 1.9 % for rice due to atmospheric nitrogen deposition applied beginning 30 days before the planting date (Fig. 7). The largest impact of nitrogen deposition is for wheat in China, which receives large amounts of nitrogen through deposition and is highly sensitive to nitrogen stress (Fig. 1). Nitrogen deposited to the surface accumulates in the soil throughout the growing season, moving quickly to lower levels of the soil profile. When fertilizer applied toward the beginning of the growing season runs out, this additional nitrogen reservoir from deposition allows for a mitigation of nitrogen stress and furthers plant growth. The fate of nitrogen in soil is not well constrained, and the length of time nitrogen is retained in the soil and useful to the plant is uncertain. The regional impacts of nitrogen deposition on crop production for this scenario are shown in Fig. 4. In all cases except for Indian rice, the nitrogen deposition effect is smaller proportionally than the enhancing effect of PM (disregarding the European maize simulation, which is restricted by kernel density).

When we apply nitrogen deposition to pDSSAT at the onset of the growing season, rather than 30 days prior to planting, we find that the impact of nitrogen deposition is dampened somewhat (production enhancement due to N deposition is then 1.6 % for maize, 1.5 % for wheat, and 1.5 % for rice). Conversely, applying nitrogen deposition in the crop model earlier enhances the increase in crop production.

If water stress is removed, the proportional enhancement of nitrogen deposition on crop production is slightly higher, as shown in Fig. 7. The largest change is for wheat, which is more water stressed than maize and rice in the model globally. While we do apply reduced and oxidized nitrogen deposition ($NH_x$ vs. $NO_y$) separately in our simulations, this separation has little impact on the above results as soil nitrification quickly converts all soil $NH_4^+$ into $NO_3^-$ in the pDSSAT model.

## 5 Conclusions

To our knowledge, this is the first effort to integrate atmospheric air quality inputs into the dynamic simulation of a crop model. Using restrictions on water and nitrogen availability and physiological limitations from the crop model provides a more realistic estimate of the impact of PM on crop production than in our earlier work which considered no such restrictions (Schiferl and Heald, 2017). Maize production increases by only 2.3 % due to PM (11.5 % in Schiferl and Heald (2017)) using the max ΔRUE = 100 % relationship, while wheat increases by 11.0 % (16.4 %) and rice increases by 3.4 % (8.9 %). The positive effect of PM on crop production is lessened when considering realistic restrictions to crop growth, but remains significant throughout the globe, especially in northern China, and may be important to consider for air quality policy decisions which may reduce PM and thereby reduce crop production. However, we note that these results assume the maximum sensitivity of crops to PM,



and more laboratory work is needed to constrain to understand how different crop varietals respond to changes in radiation throughout the growing season.

The coupling with a crop model also provides an opportunity to explore the impact of atmospheric nitrogen deposition on crop production. We find that the impact of nitrogen deposition on crop production is significant, but more modest than the effect of PM. However, a future with enhanced fertilizer inputs to feed growing populations will increase nitrogen inputs through deposition as well, potentially enhancing this effect. At the same time, lower future $NO_x$ emissions are likely due to regulatory efforts, which will reduce the nitrogen deposition flux. These reductions could also reduce PM in areas prone to ammonium nitrate formation. The future trajectory of nitrogen deposition and PM remain uncertain, and thus the net impact on global crop production is unclear.

The crop model responses to DF and nitrogen deposition examined in this study are uncertain. More work is needed, particularly controlled laboratory studies, to understand and evaluate these responses. It is critical to develop realistic crop models with reliable sensitivity to environmental factors to understand the pressure on future food security.

## Acknowledgments

Funding for this research was provided by the Martin Family Fellowship for Sustainability and the Abdul Latif Jameel World Water and Food Security Lab (J-WAFS) at the Massachusetts Institute of Technology (MIT). The authors thank the GEOS-Chem support staff and community for model documentation.

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





**Figure 1.** (top row) Crop production from base pDSSAT scenario (GC-RT SW only, no PM) with no stress applied for growing season ending in 2010. Difference in crop production due to (second row) water stress, (third row) nitrogen stress, and (bottom row) both water and nitrogen stresses. For each row: (left column) maize, (middle column) wheat, and (right column) rice. Filtered for GAEZ base crop production greater than 0.01 Mg km$^{-2}$. Global production (top) or relative production change (second row-bottom) shown in upper right.





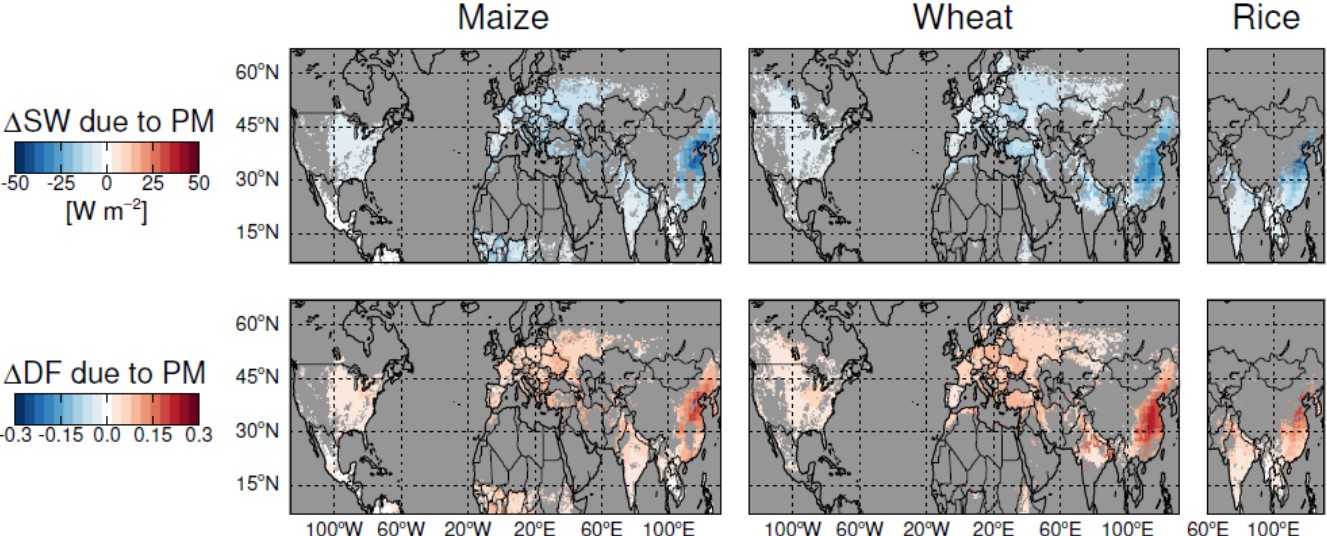

**Figure 2. Mean change in daytime (SW > 0) (top row) downward SW radiation and (bottom row) DF of the SW radiation at the surface due to PM from GC-RT. For pDSSAT growing season ending in 2010 for (left column) maize, (middle column) wheat, and (right column) rice. Filtered for GAEZ base crop production greater than 0.01 Mg km$^{-2}$.**

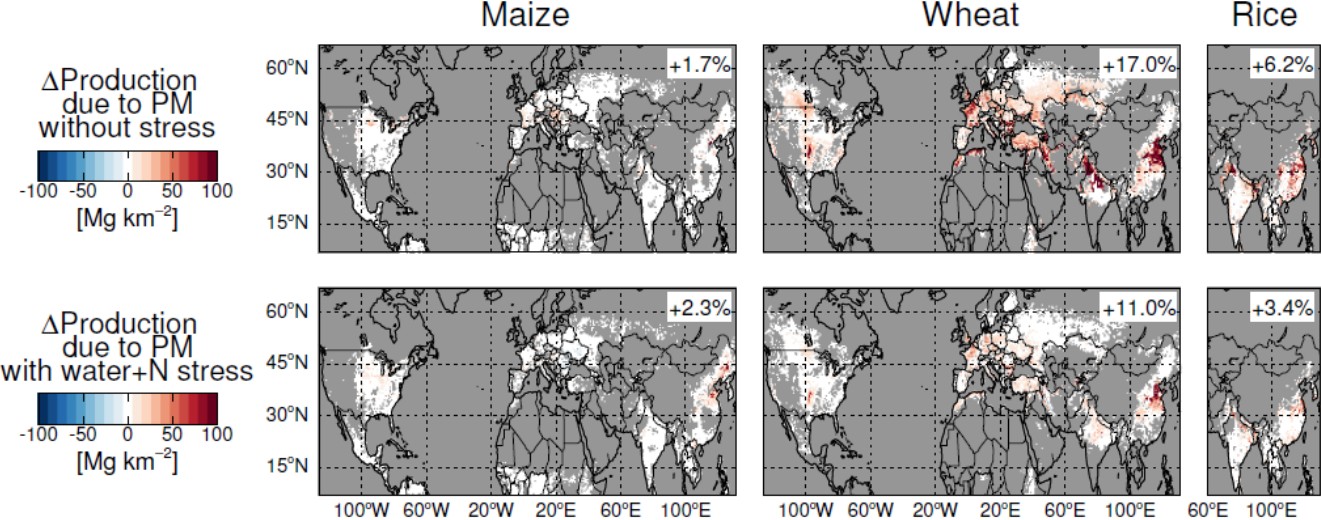

**Figure 3. Change in pDSSAT crop production due to PM with max $\Delta$RUE = 100 % with (top row) no stress and (bottom row) water and nitrogen stresses applied. For growing season ending in 2010 for (left column) maize, (middle column) wheat, and (right column) rice. Filtered for GAEZ base crop production greater than 0.01 Mg km$^{-2}$. Global relative production change shown in upper right.**



**Figure 4. Regional relative change in crop production due to PM with max ΔRUE = 100 %: offline analysis from Schiferl and Heald (2017) (blue bars with hatching), pDSSAT simulation with no stress (dark blue bars), and pDSSAT simulation with water and nitrogen stresses (light blue bars). Change due to nitrogen deposition in orange. For growing season ending in 2010 for (top row) maize, (middle row) wheat, and (bottom row) rice. Regions with a base production lower than 5 % of the global total are not shown.**





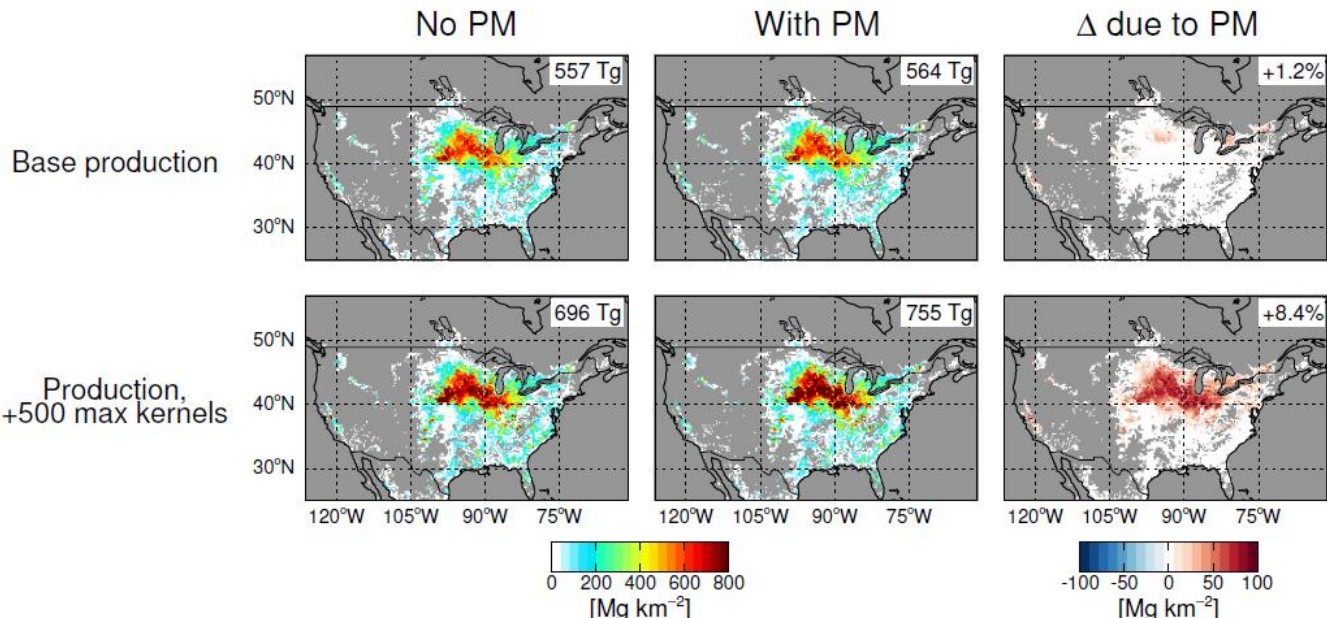

**Figure 5.** For (top row) pDSSAT base production and (bottom row) production with increased maximum kernels per plant under no stress: (left column) maize production with no PM, middle column) maize production with PM (with max $\Delta$RUE = 100 %), and (right column) maize production due to PM. Filtered for GAEZ base crop production greater than 0.01 Mg km$^{-2}$. Global production (left and middle columns) or relative production change (right column) shown in upper right.

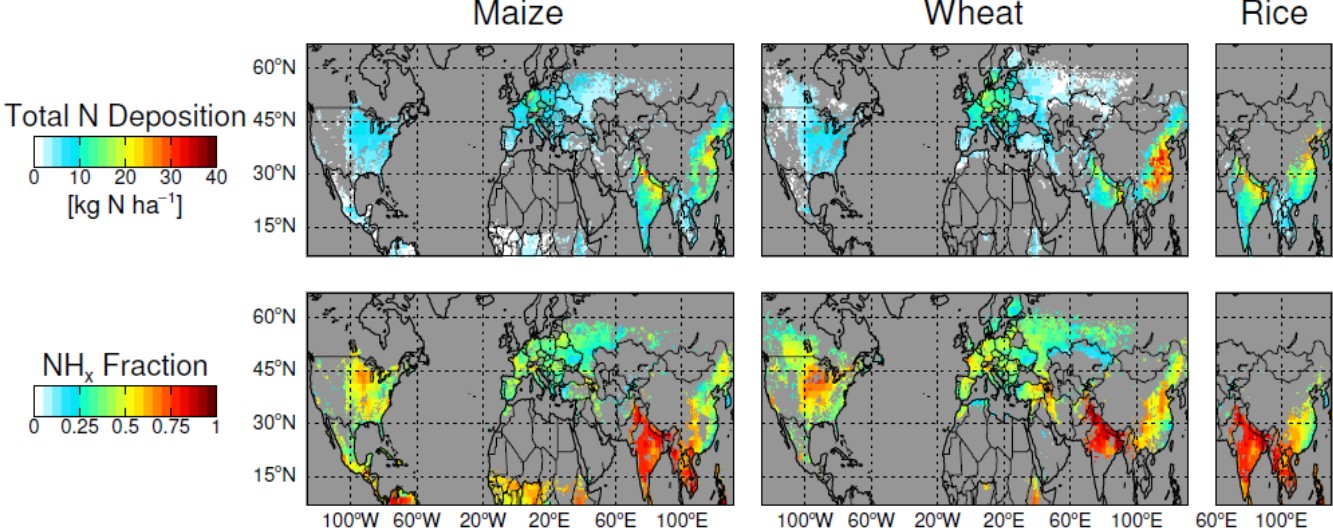

**Figure 6.** (top row) Total nitrogen deposition from GEOS-Chem and (bottom row) reduced nitrogen (NH$_x$) fraction of this total. For pDSSAT growing season ending in 2010 for (left column) maize, (middle column) wheat, and (right column) rice. Filtered for GAEZ base crop production greater than 0.01 Mg km$^{-2}$.





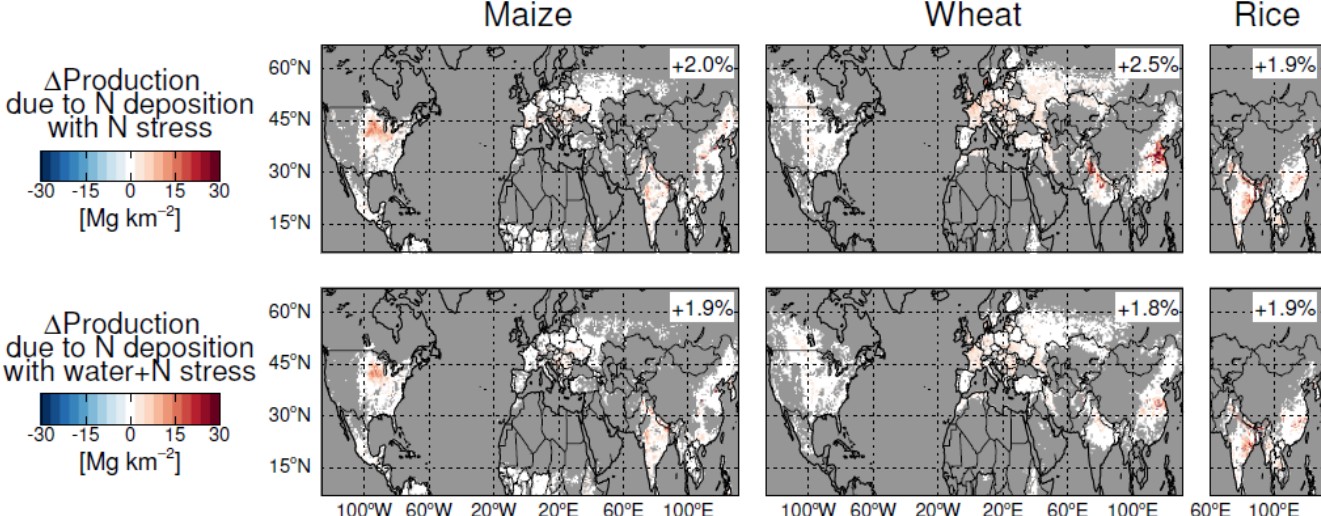

**Figure 7. Change in pDSSAT crop production due to nitrogen deposition with (top row) nitrogen stress and (bottom row) water and nitrogen stresses applied. For growing season ending in 2010 for (left column) maize, (middle column) wheat, and (right column) rice. Filtered for GAEZ base crop production greater than 0.01 Mg km$^{-2}$. Global relative production change shown in upper right.**