# Peer review of "Resource and physiological constraints on global crop production enhancements from atmospheric particulate matter and nitrogen deposition"

_Biogeosciences, 2018_

## Referee Comment (RC1) · Anonymous Referee #1 · 12 Apr 2018

General Comments: The goals of this paper are to answer questions about the impact of PM and N deposition on crop productivity, and specifically, how water, nitrogen, and plant physiological limitations contribute to the impact that PM and N deposition have on crops. These questions are important to address within the scientific community in order to increase our understanding of how anthropogenic activities affect food production. However, the introduction needs to better describe the current state of knowledge about PM and N deposition impacts on plant productively. To make the paper's arguments compelling, and to provide the reader with enough background knowledge

about the state of research on PM and N deposition impacts on plant growth, the authors need to include a much more substantial and accurate discussion of current literature. In addition, more details about the simulations and how the model handles the N cycle could be added to help the reader interpret the impact of different variables on crop production. Similarly, it would be helpful to add a discussion section or place the manuscript's results in context to other literature, including other modeling and field studies.

Specific Comments: 1. Some of the claims in the introduction are too strong, and seem to contradict the current state of research. For example, on Page 2, lines 2-3, authors say that research has been done on the impact of climate and air quality on climate without considering physiological limitations and effects of water and nutrients. However, many disciplines examine how and why changes in crop production occur with climate change or changes in resource availability, particularly with regards to water use efficiency and nitrogen use efficiency.

2. Most of the papers explaining the impacts of diffuse light on plant production are missing from the introduction, including those that discuss specific impacts on crops. For example, papers by Dev Niyogi (doi: 10.1029/2004GL020915), Kaicun Wang ( doi: 10.1029/2008GL034167, and Xiaoliang Lu (doi: 10.1016/j.agrformet.2017.02.002). Work by Gretchen Keppel-Aleks (doi: 10.1002/2016GL070052) may also be useful to discuss in this manuscript.

3. Much more literature explaining the role of N deposition on plant and crop production also needs to be included in the introduction. There is a long history of research about the different responses plants have to N deposition and why this may occur.

4. What kinds of physiological limitations/caps do the authors have in mind when they discuss this in the introduction (e.g., Rubisco/chlorophyll production, root growth, cellular physiology)?

5. Can the authors provide more details about how crop growth is calculated in

pDSSAT? Or is the crop model only a RUE model?

6. What size PM is used in the model?

7. How was the land cover and proportion of rice, wheat, and maize chosen for the crop model?

8. Page 4, Line 31: Can the authors provide more details about how the data are regridded to match the crop model resolution?

9. Page 5, Line 6: Why was PM not chosen to change evaporation, when diffuse light can also reduce air and leaf temperatures?

10. If the manuscript is discussing changes in global crop production, why do the maps in the fitures only include part of the world?

11. Page 7, Line 26: If PM includes nitrate and ammonium, how do the authors remove the confounding effect of PM on the effect of N deposition on crop productivity?

12. Figure 2: Is the mean change in daytime SW and DF a daily mean or a mean (or total reduction) in the 2010 growing season? How do the authors define growing season when different regions of the world have different growing season lengths?

13. Page 7, Line 2: By offline analysis, are the authors referring to the base simulation? If not, can they provide more details to the difference?

14. Page 7, Line 26: What was the amount of extra N added? Figure 6 shows the total amount, but it is not clear what the base case's N deposition was. How were these amounts chosen? Also, how were fertilizer amounts chosen for simulations and were these the same across the globe? Does this make it more difficult to separate out the effect of N deposition alone?

15. Page 8, Lines 15-18: Why do the authors decide to use a fertilizer application of 30 days prior to planting?

16. Page 8, Line 25: This claim that this manuscript is the first to integrate atmospheric air quality inputs into a dynamic crop model seems bold. Can the authors more specifically describe what kind of work modelers have previously done to provide context for how their own manuscript is unique? For example, there has been work done on ozone pollution on plant productivity.

17. It may also help strengthen the paper and help place the methods and results within the greater biogeochemistry discipline to include a discussion section with information about past research on the effect of N deposition and PM on modeled and/or field-based forest or crop productivity.

---

## Referee Comment (RC2) · Anonymous Referee #2 · 18 Apr 2018

This paper represents an innovative advance in the interdisciplinary research area of air pollution impacts on global food production through linking the global aerosol model (GEOS-Chem) to a crop modeling framework (pDSSAT). The study is "part 2" follow-up of a recent paper accepted for publication: Schiferl, L. D. and Heald, C. L, Particulate matter air pollution offsets ozone damage to global crop production, Atmos. Chem. Phys. Discuss., 2018. AgMIP has not traditionally assessed air pollution impacts on crops so this study is a welcome addition to the literature. The paper is brilliantly written and all figures and graphs are suitable for Biogeosciences.

[Figure]

1. There is already extensive integrated research on air pollution impacts on crops and plants. For example, flux-based risk maps e.g. Mills et al., Global Change Biology, 2011. Much of this line of work already integrates physiological constraints holistically.

2. It is difficult to understand how a crop model that does not consider water stress can be useful, or even exist in the published literature? Why was pDSSAT chosen for the study? There are process-based crop models available e.g. GLAM. Moreover, international process-based vegetation models, for instance CLM and JULES, now include crop functional types that will inherently account for water limitation. Perhaps I have misunderstood something here. Crops are very sensitive to water availability.

3. Following from point (2), it is concerning that the first paper did not include any limitation on the diffuse radiation fertilization of crops. For example, any process-based model that incorporates Farquhar-Ball-Berry photosynthesis-stomatal conductance equations will automatically include limitations on the diffuse radiation fertilization e.g. Yue and Unger, Aerosol optical depth thresholds as a tool to assess diffuse radiation fertilization of the land carbon uptake in China, ACP, 2017.

4. The aerosol and crop models have very different horizontal resolution. The aerosol model is fairly coarse (2degx2.5deg). How is the aerosol radiation output downscaled for the high resolution crop model?

5. How does pDSSAT calculate growth, biomass and yield from the potential carbon?

6. It appears that only the direct radiation changes of aerosols are calculated in GC_RT? How do aerosol indirect effects (aerosol-cloud interactions) influence the results and conclusions? Aerosol indirect radiative effects are widely accepted to be larger than the direct effects.

7. The paper is lacking in any observational evidence and as such is a theoretical modeling study. Other studies have attempted to present observational evidence e.g. Strada and Unger, Observed aerosol-induced radiative effect on plant productivity in

the eastern United States, Atmos Env 2015 and references therein; Strada and Unger, Potential sensitivity of photosynthesis and isoprene emission to direct radiative effects of atmospheric aerosol pollution, ACP, 2016; Yue et al., Future inhibition of ecosystem productivity by increasing wildfire pollution over boreal North America, ACP, 2017. One basic question is: do agricultural workers report that crops grow better and yield is higher under hazy and cloudy conditions?

8. Aerosol radiation changes will impact meteorology and hydrology (esp. temperature and soil moisture). These impacts likely have a much larger effect on crop productivity than the light changes e.g.: Yue et al., Ozone and haze pollution weakens net primary productivity in China, ACP, 2017. Is it possible that the light change impacts are over-estimated in the current approach? For instance, an 11% enhancement in global wheat would have major impacts on global economics.

---

## Short Comment (SC1) · 30 Apr 2018

*A note upfront from the submitting person: This review was prepared by Miriam Steinmann & Michelle Giust, two master students in geography at the University of Zurich. The review was part of an exercise during a second semester master level seminar on "the biogeochemistry of plant-soil systems in a changing world", which I organize. We would like to highlight that the depth of scientific knowledge and technical understanding of these reviewers represents that of master students. We enjoyed discussing the manuscript in the seminar, and hope that our comments will be helpful for the authors.*

[Figure]

General comments: This comment was written when other comments were not available yet.

The goal of this study is to analyse the effects induced by an increase of atmospheric particulate matter (PM) and nitrogen (N) deposition on crop productivity on a global scale. Both these factors are expected to increase in future due to climate change. While there have already been studies on the effect of air quality on crop production, this study is interesting because it includes environmental (water and N stress) and physiological constraints which are responsible for a dampened crop response and that have been neglected before. The authors added these new variables to an already existing model that combines global gases and particles concentrations, and meteorological data, along with soil and management informations.

Generally, the research questions are reasonable and important to address and the paper is well structured. Nonetheless, little pre-existing literature is included in the introduction and some parts of the paper. In addition, how the authors came to the results is not so clear to us yet. A short procedural part that explains the study approach in the introduction would for sure simplify the understanding for other readers that are not experts in the field, as we are. Confusing was also the fact that no discussion section was in between the results and conclusion section. In our opinion, the results could be more thoroughly discussed, they seem to just be presented but not explained or questioned. The reason for the spatial patterns of different rate of crop productivity is missing. The authors already wrote a paper in 2017 analysing the crop production of 2010, however they didn't consider physical and biological limitation. This new study tries to fill these gaps by including them in the model, therefore it refers to 2010 also. Since the analysed variables change year to year (e.g. water stress, which depends on precipitation (Page 6, Lines 7-8)), the authors should not make general assumption. Maybe re-formulate conclusion to include that all findings refer to 2010 and can differ for other years.

Specific comments:

1. Page 1, Lines 26-28: If the global decline in crop yield is considered, shouldn't localized changes in factors be included in this globality and thus get evened out?

2. Page 2, Line 1: What do the authors mean with physiological limitations? Are only the numbers of kernels per plant considered? If yes, say so in the introduction.

3. Page 2, Line 4: Is PM as abbreviation for atmospheric particulate matter commonly done, or why is the "atmospheric" not included in this abbreviation?

4. Page 2, Lines 21-24: Sentences "By artificially fixing inert nitrogen gas [. . .]" and "Anthropogenic influences on these fluxes [. . .]" are very similar, the second one could be left out.

5. Page 2 onwards: Abbreviation N for nitrogen introduced on line 21, still nitrogen is very often fully written in the following pages, but not always. Consistency?

6. Page 2, Lines 26-27: It's difficult for me to imagine that there are lot of N-poor regions in the world. Which areas are N-poor? Which not? Do these areas show different results in regard to change in productivity? Or does having too much N also count as N stress?

7. Pages 3-5: The explanation of the models used is confusing. The section includes many acronyms, sometimes explained other times not. We think the rule "less is more" is true for this section. The information should be available if a reader wants it, but should be written in a simplified way in the paper to get a clear idea of the approach used.

8. Page 4, Lines 5-6: It is unclear to us what is meant with v4.6 and v2.0, is their meaning considered common knowledge? If not, please explain

9. Page 5, Lines 2-3: We wonder why the equation for the potential carbon production for maize and wheat is different in respect to the equation for rice. An explanation should be added.
10. Page 5, Lines 25-26: It seems confusing to focus the figures on the northern hemisphere but still include global values. Either show a global map or add a sentence about why you only focus on the northern hemisphere.

11. Page 6, Line 31: What is entailed in the offline relativistic methodology? Add short explanation. Also why is it referred to here instead of the base simulation?

12. Page 7, Lines 4-6: The increase in maize production is much lower with PM in respect to rice or wheat. A reason for that seem to be the number of kernels. Then why for rice and wheat not? Why only US considered? Where is the literature? Why add 500 extra kernels and to which plant (550 kernels vs. 900 kernels) are they added? Why 500, seems arbitrary? Also, results are never discussed again and not included in the conclusion, while the other parameters (water & N stress) are compared.

13. Page 7, Lines 23-24: "[. . .], production due to also PH increases, [. . .]" bad syntax, rewrite

14. Page 7, Line 27: Why are no PM effects considered in relation to N deposition? Wouldn't this comparison make sense as both effects are found together and connected to each other in reality?

15. Page 8, Lines 15-19: How were these 30 days decided upon if the duration of deposition has such a large impact on crop production? Especially as it is stated in lines 10-11 that "[. . .] the length of time nitrogen is retained in the soil and useful to the plant is uncertain.", making an explanation of the chosen time period even more important. Unclear, elaborate.

16. all figures except Fig. 4: Color scheme unsuitable for black/white printing, maybe use white background (maps) and grayscale for scale? Also the numbers and description are too small and too close to the scale to be well legible.

17. Figure 1: Are the 2 stresses separately really necessary? They are not considered separately again later? Please avoid rainbow colour scales in your figures. See

http://rdcu.be/dWCF for a brief discussion of problems associated with these scales, and for suggestions for alternatives.

18. Figure 4: The results displayed in this figure can all be found in the other figures as well, in relation to their spatial distribution, which makes more sense in this case. Is this figure needed at all? Why is the comparison done with the offline analysis and not the base simulation?

19. Figure 5: Are "No PM" and "With PM" needed, aren't the Delta-values enough? There is a bracket missing before "[...], middle column).

---

## Author Comment (AC1) · 31 May 2018

Response to Short Comment #1 by M. W. I. Schmidt

Note: page and line references mentioned in author changes refer to positions within the revised manuscript below.

*A note upfront from the submitting person: This review was prepared by Miriam Steinmann & Michelle Giust, two master students in geography at the University of Zurich. The review was part of an exercise during a second semester master level seminar on "the biogeochemistry of plant-soil systems in a changing world", which I organize. We would like to highlight that the depth of scientific knowledge and technical understanding of these reviewers represents that of master students. We enjoyed discussing the manuscript in the seminar, and hope that our comments will be helpful for the authors.*

Generally, the research questions are reasonable and important to address and the paper is well structured. Nonetheless, little pre-existing literature is included in the introduction and some parts of the paper. In addition, how the authors came to the results is not so clear to us yet. A short procedural part that explains the study approach in the introduction would for sure simplify the understanding for other readers that are not experts in the field, as we are. Confusing was also the fact that no discussion section was in between the results and conclusion section. In our opinion, the results could be more thoroughly discussed, they seem to just be presented but not explained or questioned. The reason for the spatial patterns of different rate of crop productivity is missing. The authors already wrote a paper in 2017 analysing the crop production of 2010, however they didn't consider physical and biological limitation. This new study tries to fill these gaps by including them in the model, therefore it refers to 2010 also. Since the analysed variables change year to year (e.g. water stress, which depends on precipitation (Page 6, Lines 7-8)), the authors should not make general assumption. Maybe re-formulate conclusion to include that all findings refer to 2010 and can differ for other years.

**We thank the master students for their helpful comments. We have included additional literature in the introduction, increased context for the discussion of our results, and acknowledged the possibility of variation from year to year in the conclusions. Further, we have incorporated suggestions from several specific comments listed below.**

Specific comments:
1. Page 1, Lines 26-28: If the global decline in crop yield is considered, shouldn't localized changes in factors be included in this globality and thus get evened out?

**There could still be a net global change even if local changes of differing signs even out.**

2. Page 2, Line 1: What do the authors mean with physiological limitations? Are only the numbers of kernels per plant considered? If yes, say so in the introduction.

**The physiological limitations we have in mind include larger scale processes, such as the rate and magnitude of carbon allocation to plant carbon pools (roots, leaves, stems, grain) and life cycle timing. Cellular physiology is not represented in the crop model. This has been added on page 2, line 5.**

3. Page 2, Line 4: Is PM as abbreviation for atmospheric particulate matter commonly done, or why is the "atmospheric" not included in this abbreviation?

**Yes, PM is a standard abbreviation for atmospheric particulate matter in the atmospheric chemistry and air pollution communities.**

4. Page 2, Lines 21-24: Sentences "By artificially fixing inert nitrogen gas [. . .]" and "Anthropogenic influences on these fluxes [. . .]" are very similar, the second one could be left out.

**The first sentence refers to the fluxes of nitrogen, the second sentence refers to the balance of nitrogen within each earth system segment (e.g., land and ocean). We have improved the clarity of this on page 3, lines 23 and 25.**

5. Page 2 onwards: Abbreviation N for nitrogen introduced on line 21, still nitrogen is very often fully written in the following pages, but not always. Consistency?

**We have updated the consistency. Although introduced for reference as N on page 3, line 22, nitrogen is now fully written in all cases throughout the text. N is still used in the figure captions to be consistent with figure labeling, in which abbreviation is necessary for space considerations.**

6. Page 2, Lines 26-27: It's difficult for me to imagine that there are lot of N-poor regions in the world. Which areas are N-poor? Which not? Do these areas show different results in regard to change in productivity? Or does having too much N also count as N stress?

**Developing regions including Africa, South America and India, as well as parts of Eastern Europe could be considered nitrogen-poor, defined by the potential to increase yield with additional nitrogen input (Mueller et al., 2012). This has been added on page 4, lines 14-15. Industrial regions which can afford to purchase fertilizer such as in the United States, Western Europe and China are generally not nitrogen-poor, and in some cases apply too much nitrogen. While this excess nitrogen negatively affects other segments of the environment, it is not considered nitrogen stress in pDSSAT. The nitrogen-poor regions listed above do not entirely correspond to the regions affected by nitrogen stress in Fig. 1 because this figure shows total production, rather than yield. Regions with large crop areas will magnify the apparent impact of nitrogen stress on yield. This is noted on page 9, lines 16-17.**

7. Pages 3-5: The explanation of the models used is confusing. The section includes many acronyms, sometimes explained other times not. We think the rule "less is more" is true for this section. The information should be available if a reader wants it, but should be written in a simplified way in the paper to get a clear idea of the approach used.

**We think that this particular comment reflects the non-expert status of the reader. This style of model description is needed for those who aim to understand the model inputs and wish to reproduce the setup used in our study.**

8. Page 4, Lines 5-6: It is unclear to us what is meant with v4.6 and v2.0, is their meaning considered common knowledge? If not, please explain

**As these models change over time, we identify the specific model versions used in our study. The "v" nomenclature for this is commonly used. We have modified their usage slightly for clarity on page 6, lines 14-15.**

9. Page 5, Lines 2-3: We wonder why the equation for the potential carbon production for maize and wheat is different in respect to the equation for rice. An explanation should be added.

**These are the relationships parameterized in the crop-specific CERES growth modules. This is added in page 7, line 27.**

10. Page 5, Lines 25-26: It seems confusing to focus the figures on the northern hemisphere but still include global values. Either show a global map or add a sentence about why you only focus on the northern hemisphere.

**We focus on the significantly productive areas of maize, wheat, and rice for clarity of the figures. This is described on page 8, lines 28-30 and is consistent with our earlier work (Schiferl and Heald, 2018).**

11. Page 6, Line 31: What is entailed in the offline relativistic methodology? Add short explanation. Also why is it referred to here instead of the base simulation?

**We have added clarifications to the text on page 10, lines 3-5.**

12. Page 7, Lines 4-6: The increase in maize production is much lower with PM in respect to rice or wheat. A reason for that seem to be the number of kernels. Then why for rice and wheat not? Why only US considered? Where is the literature? Why add 500 extra kernels and to which plant (550 kernels vs. 900 kernels) are they added? Why 500, seems arbitrary? Also, results are never discussed again and not included in the conclusion, while the other parameters (water & N stress) are compared.

**Rice and wheat production increase substantially when the PM effect is applied. Thus, they are not being restricted by the same mechanism as maize, which shows much less enhancement. Focusing on the US and imposing this arbitrary increase in kernel capacity is used as an example, as most of the global production occurs there. We have included clarifications throughout the paragraph starting on page 10, line 14. The importance of accounting for physiological restrictions is mentioned again in the conclusions starting on page 12, line 17 and explains the significant difference between the PM effect result presented here and in the offline analysis (Schiferl and Heald, 2018).**

13. Page 7, Lines 23-24: "[. . .], production due to also PH increases, [. . .]" bad syntax, rewrite

**Corrected.**

14. Page 7, Line 27: Why are no PM effects considered in relation to N deposition? Wouldn't this comparison make sense as both effects are found together and connected to each other in reality?

**We have clarified that nitrogen-containing PM deposition is taken into account here, but we do isolate the nitrogen deposition impacts from the PM light impacts to separately quantify these.**

15. Page 8, Lines 15-19: How were these 30 days decided upon if the duration of deposition has such a large impact on crop production? Especially as it is stated in lines 10-11 that "[. . .] the length of time nitrogen is retained in the soil and useful to the plant is uncertain.", making an explanation of the chosen time period even more important. Unclear, elaborate.

**We have added acknowledgment that 30 days is an arbitrary timeframe on page 8, line 15. We now mention a possible method for better constraining this timeframe on page 12, lines 1-3.**

16. all figures except Fig. 4: Color scheme unsuitable for black/white printing, maybe use white background (maps) and grayscale for scale? Also the numbers and description are too small and too close to the scale to be well legible.

**Optimizing figures for black and white can infringe on clarity; given that this journal is online open-access, all readers will have access to color versions. Furthermore, we maintain consistency in style between the figures in this paper and our previous work (Schiferl and Heald, 2018). The gray map vs. white 0-values is needed to identify croplands.**

17. Figure 1: Are the 2 stresses separately really necessary? They are not considered separately again later? Please avoid rainbow colour scales in your figures. See http://rdcu.be/dWCF for a brief discussion of problems associated with these scales, and for suggestions for alternatives.

**Yes, they are needed separately as the areas impacted by nitrogen stress in the base simulation will be more sensitive to nitrogen deposition (page 11, line 22). There is considerable debate over color scales and differing opinions on how these should be used (the link suggested by the students is in fact a commentary not a scientific study). While we agree that rainbow color scales are not always appropriate, given that the objective of our figures is to highlight the large-scale differences around the world, this color scale is ideal. We also choose these color scales to maintain consistency in style between the figures in this paper and our previous work (Schiferl and Heald, 2018).**

18. Figure 4: The results displayed in this figure can all be found in the other figures as well, in relation to their spatial distribution, which makes more sense in this case. Is this figure needed at all? Why is the comparison done with the offline analysis and not the base simulation?

**We find it important to clearly identify the regional differences (and consistencies) in these impacts conveyed through Fig. 4. The filled bars depict the changes which occur in cases due to PM or nitrogen deposition. The base simulation is used in the calculation of the change due to nitrogen deposition (page 11, line 6). The offline analysis is most comparable to the first case – impact of PM with no stress.**

19. Figure 5: Are "No PM" and "With PM" needed, aren't the Delta-values enough?
There is a bracket missing before "[. . .], middle column).

**It is important to show both the delta-values and the magnitudes in each scenario because the "no PM" values under base production and +500 max kernels also change. The figure illustrates the restriction places on maize growth even without PM. We corrected the missing bracket in the caption.**

**Resource and physiological constraints on global crop production enhancements from atmospheric particulate matter and nitrogen deposition**

Luke D. Schiferl[1], Colette L. Heald[1,2], and David Kelly[3]

[1]Department of Civil and Environmental Engineering, Massachusetts Institute of Technology, Cambridge, Massachusetts, USA

[2]Department of Earth, Atmospheric and Planetary Sciences, Massachusetts Institute of Technology, Cambridge, Massachusetts, USA

[3]University of Chicago Computation Institute, Chicago, IL, USA

*Correspondence to*: Luke D. Schiferl (schiferl@mit.edu)

**Abstract.** Changing atmospheric composition, induced primarily by industrialization and climate change, can impact plant health and may have implications for global food security. Atmospheric particulate matter (PM) can enhance crop production through the redistribution of light from sunlight to shaded leaves. Nitrogen transported through the atmosphere can also increase crop production when deposited onto cropland by reducing nutrient limitations in these areas. We employ a crop model (pDSSAT), coupled to input from an atmospheric chemistry model (GEOS-Chem), to estimate the impact of PM and nitrogen deposition on crop production. In particular, the crop model considers the resource and physiological restrictions to enhancements in growth from these atmospheric inputs. We find that the global enhancement in crop production due to PM in 2010 under the most realistic scenario is 2.3 %, 11.0 %, and 3.4 % for maize, wheat, and rice, respectively. These crop enhancements are smaller than those previously found when resource restrictions were not accounted for. Using the same model setup, we assess the effect of nitrogen deposition on crops and find modest increases (~2 % in global production for all three crops). This study highlights the need for better observations of the impacts of PM on crop growth and the cycling of nitrogen throughout the plant-soil system to reduce uncertainty in these interactions.

**1 Introduction**

Population growth is intensifying stress on global food production. Simultaneously, anthropogenic activities are changing many aspects of the earth system. This reinforces the need to better understand how crop production may be affected by changes to the water, air, light, and soil required for efficient growth. For example, Challinor et al. (2014) suggest a global decline in crop yield due to climate change of more than 10 % is likely by 2050. This is uncertain, however, and the projected sign and magnitude varies by crop and region due to localized changes in factors such as temperature and precipitation combined with global carbon dioxide ($CO_2$) enhancement (IPCC, 2014). Many studies have explored the impacts of climate and air quality on crop production, both globally and regionally, with various results

depending on the tools, methods and processes used by each (e.g., Burney and Ramanathan, 2014; Lobell and Burke, 2010; Shindell et al., 2011; Tai et al., 2014). Investigations of the impacts of air quality on crops, in particular, have focused mainly on the negative impact of ozone pollution (Avnery et al., 2011; Mills et al., 2011; Van Dingenen et al., 2009). In comparison, only limited work has been conducted to assess how atmospheric particulate matter (PM) impacts crop production, and this

5 has  been done without considering physiological limitations (e.g., rate and magnitude of carbon pool allocation) and other environmental stresses (e.g., water and nutrients) (Greenwald et al., 2006; Schiferl and Heald, 2018).

Emitted through combustion and natural processes and formed through chemical

10 oxidation in the atmosphere, PM is the leading cause of air quality issues globally and is responsible for over 4 million premature deaths per year (Cohen et al., 2017). PM also impacts crop production by modifying shortwave radiation reaching the surface. Through the scattering of light, PM decreases the total shortwave (SW) radiation at the Earth's surface, which is made up of direct and diffuse light (SW = direct + diffuse). PM also increases the diffuse fraction (DF) of this SW radiation $(DF = \frac{\text{diffuse}}{\text{SW}}$$)$. Increased DF more evenly distributes light throughout the canopy of a plant, redirecting light away from (at

15 times over-saturated) leaves in direct sunlight and onto shaded leaves. In this way, plants can more efficiently make use of incoming solar radiation (Kanniah et al., 2012).

Forests dominate previous studies of the impact of PM on plant productivity. Using network observations, Niyogi et al. (2004) showed that the $CO_2$ sink, a measure of plant productivity, increases with PM, indicated by aerosol optical depth (AOD), over

20 forests, but decreases for grasslands. More recent work related satellite AOD measurements to observations from $CO_2$ flux towers to quantify the impact of diffuse light on plant productivity. In the Amazon forest, enhancements in net ecosystem exchange (NEE) of up to 29 % are observed when the DF reaches approximately 0.5 (Cirino et al., 2014). Strada et al. (2015) find an increase in midday gross primary productivity (GPP) of ~13 % in US deciduous forests when DF is 0.4–0.6. Advanced canopy or leaf-scale process modelling has been used to further examine how PM impacts natural vegetation and the carbon

25 cycle. The model framework of Strada and Unger (2016) shows little sensitivity in global total GPP (~1–2 %) to PM pollution, with regional enhancements of ~5–8 % in North America and Eurasia and ~2 % in the Amazon, where forested canopies dominate. In China, Yue and Unger (2017) use AOD thresholds along with satellite observations and vegetation modelling to find the impact of PM pollution on net primary production (NPP) varies spatially from –3 % to +6 %. When accounting for the direct impacts of PM on light, temperatures, and hydrology, Yue et al. (2017) find a net increase in NPP of 5 %.

The impact of PM on managed vegetation (crops) is less well studied than for natural vegetation. PM can increase growth and production of crops when the increase in efficiency outweighs the loss of SW radiation. This depends on the local light conditions (changes in SW vs. DF) and crop type ($C_3$ vs. $C_4$). $C_4$ crops such as maize are less likely to be light

saturated than $C_3$ crops such as wheat. Niyogi et al. (2004) find that the $CO_2$ sink increases over croplands. In contrast with their forest sites, Strada et al. (2015) find a decrease in midday GPP of ~17 % associated with high observed AOD for a combination of US croplands and grasslands sites. They attribute this difference to canopy architecture which minimizes leaf shading, and thus the impact of diffuse light, when the sun is overhead. Greenwald et al. (2006) use relationships between DF, determined by climatological AOD, and a crop's radiative use efficiency (RUE), a measure of how effective a plant converts light into carbon, from Sinclair et al. (1992) along with varying meteorology and a crop model to estimate the impact of PM on crop yield. Assuming no restrictions on growth due to stresses at several locationssites, they find a large variation in impacts based on the DF-to-ΔRUE relationship chosen. Under the maximum relationship, maize increases by 0–10 %, wheat increases by 0–5 % and rice increases by 0–40 % under varying cloud conditions (Greenwald et al., 2006). Using this approach, but with a combined atmospheric chemistry and radiative transfer model to better simulation spatial and temporal variability of PM impacts on radiation, Schiferl and Heald (2018)Schiferl and Heald (2017) estimate a global positive impact of PM of 12 %, 16 %, and 9 % on maize, wheat, and rice production, respectively, for the year 2010. While their study uses a simple representation of the PM impacts on crop productivity, their approach isolates the impact of PM on crop production, which is not easily estimated based on previous observational analyses or mechanistic models. Observed AOD impacts on radiation are convolved with the influence of clouds, and as it is difficult to isolate only PM impacts, we cannot easily translate the observed AOD-to-carbon flux relationships to the impacts of DF on RUE. Mechanistic model studies account for all land biomass, but while crops do have smaller canopies and are less sensitive to DF then trees, such models do not differentiate between individual crop characteristics (e.g., canopies, growing seasons). Furthermore, the observed and simulated changes in NEE and GPP in these studies do not correspond directly to crop production, but rather on carbon uptake or release.

Industrial agriculture, driven by the need to produce food for a growing human population, has modified the global nitrogen (N) cycle (e.g., Bouwman et al., 2013; Smil, 1999). By artificially fixing inert nitrogen gas into reactive forms, humans have increased the fluxes of nitrogen throughout the environment, including into the atmosphere, onto land, and into the water (Galloway and Cowling, 2002). Nitrogen species in the atmosphere, both reduced and oxidized, return to the surface through deposition processes after being transported away from source regions. Anthropogenic influences on this deposition ese fluxes change the nitrogen balance in land and water ecosystems. In natural systems, this can cause acidification and eutrophication, which negatively impacts the biosphere (Beem et al., 2010; Erisman et al., 2007). Nitrogen accumulation into ecosystems from deposition is the main driver leading to an overall reduction in biodiversity, and while other factors such as direct toxicity, soil acidification, and increased susceptibility to stress are secondary, they can be dominant locally (Bobbink et al., 2010). While remaining substantially higher than during preindustrial time, current rates of nitrogen deposition have recently declined over the US and Europe but are expected to increase in developing countries in the future (Lamarque et al., 2013). This will contribute to projected (for 2050) nitrogen surpluses in Africa and Latin America corresponding with increases in crop and livestock production (Bouwman et al., 2013).

By including the coupling of carbon and nitrogen in a land-surface model, Thornton et al. (2007) show that GPP is limited by the supply of nitrogen to the biosphere and simulated over 40 % less GPP than a case that does not include this limitation. This carbon-nitrogen coupling dampens the response of vegetation to $CO_2$ concentration increases by over 70 %. The addition of atmospheric nitrogen deposition in the coupled system increases global GPP by ~2 %. When integrated into a fully-coupled

5    earth system model, there is a decrease in carbon uptake from $CO_2$ fertilization and an increase in carbon uptake from climate warming from the interactions between carbon and nitrogen. This increase in carbon uptake is due to enhanced nitrogen mineralization in the soil from a higher rate of decomposition (Thornton et al., 2009). Thomas et al. (2013) show that these simulated carbon-nitrogen responses for forests are smaller than those observed. Their model modifications result in greater retention of nitrogen deposition in biomass and a tighter coupling between nitrogen deposition and rising atmospheric $CO_2$

10    concentration. The model better represents observations by increasing the aboveground carbon storage response to nitrogen deposition.

The deposited nitrogenNitrogen deposition can also impact crop production, by providing additional fertilization, increasing yields in areas which are nitrogen limited (Goulding et al., 1998). These areas include portions of Africa, South America,

15    India, and Eastern Europe (Liu et al., 2010b; Mueller et al., 2012). Liu et al. (2013) show that in China nitrogen deposition leads to increased nitrogen uptake in non-fertilized croplands, resulting in a small increase in yield (1 t ha$^{-1}$) derived from a nitrogen uptake to yield ratio. Lassaletta et al. (2014) develop relationships between observed total nitrogen input and crop yield on a countrywide basis, but they do not disaggregate the impacts of deposition saying only that the input from deposition is small, but not negligible. While Ladha et al. (2016) estimate that 613 % of nitrogenN contained in global maize, wheat, and

20    rice comes from deposited nitrogen, to date, there has been no global study of the change of yield associated with nitrogen deposition, with most studies concentrating on the impacts of nitrogen deposition on interactions with atmospheric $CO_2$ and carbon storage. Folberth et al. (2016) neglect nitrogen deposition in their study of soil and meteorological data uncertainties in crop models due to the lack of available deposition data in a form suitable for global crop models.

25    Finally, PM and nitrogen deposition are also connected: The release of excess nitrogen from fertilizer application and livestock production in the form of ammonia ($NH_3$) contributes to PM formation in the atmosphere under acidic conditions (Seinfeld and Pandis, 2006). Nitric acid ($HNO_3$), an oxidized form of nitrogen oxides ($NO_x$) emissions from mobile and industrial sources, contributes both to the nitrogen burden and these acidic conditions. Nitrogen can also be incorporated in PM as organic nitrates when biogenic volatile organic compounds (BVOCs) react with $NO_x$ (Mao et al., 2013). Recent global modeling

30    studies incorporate more complex nitrogen transformations and cycling, such as the implementation of bi-directional ammonia fluxes into atmospheric chemistry models (Zhu et al., 2015) and climate-dependant agricultural nitrogen pathways into earth system models (Riddick et al., 2016).

Schiferl and Heald (2018) quantify the impact that air quality (ozone and PM) has on current and future global crop production. Their analysis, while consistent with the approach generally applied to estimate air quality impacts on crops in previous studies mentioned above , fails to account for the set of physical and biological restrictions placed on crop growth and production. In particular, they consider crop production enhancement due to the diffuse effect of PM  to be unlimited. However, water and nitrogen stresses and physiological caps placed on crop production may dampen these responses. This study is a direct follow-up to Schiferl and Heald (2018), where we employ a crop  model to simulate the enhancements in crop production associated with PM and nitrogen deposition simulated by an atmospheric chemistry and radiative transfer model and explore the potential impact of resource and physiological constraints on this production.

**2 GEOS-Chem Atmospheric Chemistry Model**

The GEOS-Chem model (www.geos-chem.org, last access: June 2015) simulates the global concentration of gases and particles in three dimensions. Simulated PM concentrations are read into the Rapid Radiative Transfer Model for GCMs (RRTMG) to estimate the impact of PM on radiation throughout the atmosphere (Heald et al., 2014). Together these models are referred to as GC-RT. The model version and setup used here is the same as for the standard 2010 emissions scenario described by Schiferl and Heald (2018). In brief: GC-RT is run at $2° \times 2.5°$ horizontal resolution using GEOS-5 meteorology for the years 2009 and 2010 from the NASA Global Modeling and Assimilation Office (GMAO). In this study, PM refers to the sum of all simulated aerosol species: sulfate ($SO_4^{2-}$), nitrate ($NO_3^-$), ammonium ($NH_4^+$), black carbon (BC), organic carbon (OC), sea salt and dust. Inorganic aerosol thermodynamics are coupled to an ozone–VOC–$NO_x$–oxidant chemical mechanism, where ISORROPIA II (Fountoukis and Nenes, 2007) handles the gas-particle phase partitioning of ammonium nitrate. GC-RT simulates wet and dry deposition of both aerosols and gases (Amos et al., 2012; Liu et al., 2001; Wang et al., 1998; Zhang et al., 2001). Major global anthropogenic gas emissions come from the Emission Database for Global Atmospheric Research (EDGAR) v4.2 ($NO_x$, carbon monoxide (CO), sulfur dioxide ($SO_2$)) , the Reanalysis of the TROpospheric chemical composition (RETRO) inventory (non-methane VOCs) (Hu et al., 2015), and the Global Emission Inventory Activity (GEIA) inventory ($NH_3$). These are overlaid by regional inventories where available (see Schiferl and Heald (2018) for details). Additional $NO_x$ emissions are from lightning and soil, described by Murray et al. (2012) and Hudman et al. (2012), respectively. Directly emitted aerosol sources include anthropogenic black carbon  and organic carbon  (Bond et al., 2007; Leibensperger et al., 2012), dust (Fairlie et al., 2007), and sea salt (Jaeglé et al., 2011). GC-RT uses a bulk aerosol scheme, where each aerosol species is described by a fixed log-normal size distribution, the physical and optical properties of which are described in Heald et al. (2014) and are accounted for in the radiative transfer scheme. PM sizes in GC-RT span several orders of magnitude, with mean diameters that range, for example, from black carbon, 0.04 μm, to sulfate, 0.14 μm, to dust, 8 μm.

In this study, we use hourly output of surface SW radiation and the diffuse and direct portions of this SW radiation from GC-RT both with and without PM under all-sky (real time variation in cloudiness) conditions. These are used to calculate the DF of the SW radiation.  While SW and DF respond differently to the differing properties of each PM type, here we consider the net effect of all PM. The impacts of PM described in this study account only for the direct radiation changes through light absorption and scattering, and do not consider secondary feedbacks of aerosol on clouds, meteorology, and hydrology. We also use daily output of nitrogen deposition flux from the atmosphere, including the wet and dry deposition simulated for all nitrogen species. Nitrogen mass deposited from five species, ammonia, ammonium, nitric acid, nitrate, and nitrogen dioxide ($NO_2$), make up 98 % of the total simulated nitrogen deposition for 2010. Both the PM impacts on surface radiation and the nitrogen deposition flux from the atmosphere are derived from the same GC-RT simulation, providing consistency over the emissions, chemistry, and deposition schemes described above.

**3 pDSSAT Crop Model**

**3.1 Model Description**

We use  the Decision Support System for Agrotechnology Transfer (DSSAT) v4.6 crop system model (Hoogenboom et al., 2015), along with  the parallel System for Integrating Impact Models and Sectors (pSIMS) v2.0 (Elliott et al., 2014), together called pDSSAT, to simulate the global production of maize, wheat, and rice. DSSAT provides a unified interface which combines various crop simulation models (Jones et al., 2003). Inherently a point model, DSSAT uses daily meteorological data (minimum temperature, maximum temperature, precipitation, solar radiation, wind speed, and relative humidity) along with soil and management information at a given location. The model then calculates a crop yield at harvest taking into account soil-plant-atmosphere dynamics throughout the growing season. Plant growth, in our case, is determined by the Crop-Environment Resource Synthesis (CERES) model module for each crop . CERES modules, developed separately for maize, wheat, and rice, simulate the carbon and nitrogen pools, among other parameters, associated with the various plant parts (e.g., leaves, stems, roots, grain) throughout the growth stages of each crop type. Potential dry matter (carbon) production is determined as a function of the solar radiation, SW (see Eq. 1). The actual dry matter production at each time step is limited by the effects of non-optimal temperature, water stress, and/or nitrogen stress, if applicable. Water and nitrogen stresses are determined by comparing the requirements of each crop with the amount of each resource available to the plant. Dry matter produced is then distributed into the plant parts based on those associated with the growth stage at that time. The sensitivity of growth rates and physical limitations for each plant part during each growth stage is determined by the physiology of that crop and cultivar (Jones et al., 1986; Ritchie et al., 1998; Ritchie and Otter, 1985). The simulation of these individual plant parts, rather than only total carbon, is of critical importance for this study as we are concerned with the production of grain to address impacts on food security. A recent review of CERES performances for maize, wheat, and rice finds that the models reproduce observed

grain yield well, with relative errors of ~10%, ~20%, and ~10%, respectively (Basso et al., 2016). They also find that secondary parameters such as soil temperature and nitrogen cycling were much less well represented.

pSIMS allows for the globally gridded simulation of crop yield by running DSSAT in parallel at various grid boxes using consistent data and setting input methods (Elliott et al., 2014). In our study, we set pDSSAT to run at $0.5° \times 0.5°$ horizontal resolution. This is only limited by the availability of suitable global input data. pDSSAT uses daily meteorological information from AgMERRA (Ruane et al., 2015), a version of the NASA Modern-Era Retrospective Analysis for Research and Applications (MERRA) product developed for use in the Agricultural Model Intercomparison and Improvement Project (AgMIP) (Rosenzweig et al., 2013). We note that this meteorological product is closely related to the GEOS-5 product which drives the GC-RT simulations. Soil inputs come from the Global Soil Dataset for Earth System Modeling (GSDE) (Shangguan et al., 2014). Additional required information includes the range of planting dates (Portmann et al., 2010; Sacks et al., 2010), distribution of cultivars (based on local growing degree days (GDD)), and fertilizer application amounts at each grid box. We use fertilizer information from the Spatial Production Application Model (SPAM, (You et al., 2012) at each grid box. We highlight that direct fertilizer application is the only source of nitrogen supplied to crops in the pDSSAT model in addition to the baseline nitrogen content in each soil layer given by GSDE. Except for the soil inputs, which are modified in pSIMS v2.0, these pDSSAT input data listed above are consistent with those used by the global gridded crop model (GGCM) intercomparison portion of AgMIP (Rosenzweig et al., 2014).

**3.2 Integration of GEOS-Chem with pDSSAT**

Using the hourly SW, diffuse and direct radiation output from GC-RT, we calculate the daily mean daytime (SW > 0) SW and DF for each GEOS-Chem gridbox ($2° \times 2.5°$ horizontal resolution) for all of 2009 and 2010. We group the nitrogen deposition fluxes of individual species into two groups, reduced nitrogen ($NH_x$) and oxidized nitrogen ($NO_y$), and calculate the daily total flux for each group for the same time period. The daily SW and DF values, along with the daily $NH_x$ and $NO_y$ deposition flux values, are regridded to the pDSSAT grid and resolution using area-weighted regridding and integrated into the input meteorology.

For the PM simulations, the daily SW and DF are used in the pDSSAT crop-specific plant growth modules to modify the potential carbon production. Following each crop-specific CERES growth module, Eq.uation 1 is used for maize and wheat, and Eq. 2 is used for rice:

$$P_{carb} \propto 0.5 \times SW \times RUE_{s,DF} \qquad (1)$$

$$P_{carb} \propto (0.5 \times SW)^{0.65} \times RUE_{s,DF} \qquad (2)$$

where $P_{carb}$ is the potential carbon production, SW is the daily mean shortwave radiation from GC-RT, and $RUE_s$ is crop-specific radiation use efficiency (Ritchie et al., 1998). For simulations with PM affecting SW and DF, SW modified by PM from GC-RT is used as input for these relationships in Eqs. 1 and 2 only and is not used in other functions dependent on solar

radiation, such as evaporation (i.e., the GC-RT SW without PM remains applied to these processes). Schiferl and Heald (2017), to modify the RUE$_s$ based on the DF, where max ΔRUE = 100 % at DF = 0.8 (Greenwald et al., 2006). This represents the upper-limit of potential PM impacts on crop production. We note that additional processes which impact plant productivity, such as

5    evapotranspiration and water use efficiency, have also been shown in both observations and simulations to be affected by changes in DF (Lu et al., 2017; Wang et al., 2008). These second-order effects may dominate the crop response under certain conditions and therefore should be included any assessment of the overall environmental impacts on crop growth. However, the goal of this work is to explore only the direct impact of radiation changes (due to PM) on crop productivity, enabling a comparison with Schiferl and Heald (2018).

For the nitrogen deposition simulations, NH$_x$ and NO$_y$ fluxes are applied daily as an additional source of fertilizer to the surface layer of the soil as NH$_4^+$ and NO$_3^-$, respectively, due to their similar behaviors in soils (Ladha et al., 2016). We apply these deposition fluxes beginning 30 days prior to the planting date at each location. The timing of this initiation is uncertain, as the fate of deposited nitrogen is not well constrained, and the impacts of nitrogen deposition can be assessed over a single growing

15    season to multi-year time scales (Goulding et al., 1998). Our selection of 30 days is therefore somewhat arbitrary. We discuss the impact of this assumption in Sect. 4.2.

**3.3 Base Simulation**

We configure pDSSAT to run for 2009 and 2010 with water and nitrogen stress turned off. Our modification for potential carbon production using input from GC-RT is applied to SW only (with SW values from GC-RT without PM). Maize, wheat,

20    and rice are simulated independently. We sample the results for each crop for the growing season ending in 2010. For example, crops planted in northern hemisphere spring and harvested in fall are grown entirely within 2010, while winter crops are planted in fall 2009 and harvested in spring 2010. These planting and harvest dates are determined within pDSSAT by the life cycle characteristics of each crop and vary based on the location-specific meteorological (e.g., GDD, timing of rainfall) and resource (e.g., fertilizer amount, irrigated v. rainfed) inputs for that simulation. For a consistent comparison, we determine crop

25    production by multiplying the pDSSAT crop yield by the crop area from the Global Agro-Ecological Zones (GAEZ) assessment for 2000 (FOA, 2016)(www.fao.org/nr/gaez) scaled to 2010 as in Schiferl and Heald (2018), rather than by using the internal pDSSAT harvested area parameter. The results from this simulation, our base simulation, are shown in Fig. 1. Also as in Schiferl and Heald (2018), we focus our figures on the industrialized areas of the northern hemisphere, which rely heavily on maize, wheat, and rice, though all numbers presented

30    are global. Since our base simulation has no restrictions on water and nitrogen (both the nitrogen supply and irrigation are unlimited), the simulated crop production vastly surpasses that from GAEZ. For maize, this is 2062 Tg from pDSSAT compared to only 871 Tg from GAEZ. Simulated wheat production is 2591 Tg, and simulated rice production is 1250 Tg compared to GAEZ values of 667 Tg and 705 Tg, respectively.

We rerun the crop model with water stress only, nitrogen stress only, and both stresses together to characterizetest the sensitivity of the base simulation to these resources (Fig. 1). Water stress occurs when the amount of soil water available is below the potential transpiration rate of the plant. For maize, the negative effect of water stress on production is most evident in the United States (US) Plains and northern China and produces a 29 % production reduction globally. The effect of water stress is larger globally on wheat (40 % reduction), and is largest in the southern US Plains, northern China, and throughout western Asia. Rice production is impacted the least by water stress, with only a 14 % reduction in production when imposing water stress, mostly in northern India. Water stress is dependent on the precipitation prescribed from the meteorology of that growing season, so these results will vary from year to year.

Nitrogen stress occurs when the plant tissue nitrogen concentration is less than the critical nitrogen concentration determined to provide optimal growth. In our base simulation, nitrogen stress follows different patterns compared to water stress for many regions and crops, although the global magnitudes in production reduction are similar. This response to carbon-nitrogen coupling is similar in sign and magnitude as that found for global GPP by Thornton et al. (2007). Maize production is affected by nitrogen stress primarily in the US Plains and Midwest. Nitrogen stress for wheat is distributed into all regions, while the effect on rice production is again lowest globally, it is and largest in southeast (SE) Asia. We note that the apparent impact of nitrogen stress on maize in Midwestern US is magnified by the large crop area in this region. Nitrogen stress is more similar from year to year in the model as fertilizer application, which provides nitrogen to the soil, and inherent soil nitrogen content is identical for all simulation years. Small variations do exist as variable temperatures and radiation impact the onset of crop growth stages and use of nitrogen. Folberth et al. (2016) find that uncertainty in soil data can impact simulated crop yield variability more than meteorological variability, especially for no water stress (irrigated), high nitrogen stress areas. In contrast, they find that irrigated areas with high nitrogen inputs show little difference between yield due to soil and meteorological input variability. Total production change due to both water and nitrogen stress does not combine linearly. This illustrates the interconnected system simulated by the crop model. Overall, these environmental and management constraints greatly reduce global crop production from its unstressed potential. They are important to consider when analyzing the impact of PM and nitrogen deposition on crop production.

**4 Results**

**4.1 Impact of Particulate Matter on Crop Growth**

To simulate the effect of PM on crop production, we run pDSSAT as above (for 2009 and 2010, sampling to the growing season ending 2010) with SW and DF input from GC-RT with and without PM. The differences in SW and DF due to PM over the pDSSAT growing season (determined by the base simulation) are shown in Fig. 2. PM has a negative effect on SW everywhere and positive effect on DF. The largest influence of PM is over China for all three crops. The influence is especially

noticeable for wheat, where a growing season over the winter corresponds with higher PM concentrations. The difference between the simulations with and without PM is the change in production due to PM, and this is shown in Fig. 3. We perform this procedure first with no stress factors applied in order to compare to the results found in Schiferl and Heald (2018), referred to here as the "offline analysis". The offline analysis uses a relativistic methodology which allows for unlimited growth

5    enhancement (or loss) and is determined by the accumulated PM impacts throughout the growing season.  In this pDSSAT simulation with no stress applied, global maize production increases by 1.7 %, wheat increases by 17.0 %, and rice increases by 6.2 %. Wheat production in the India and China+SE Asia regions is most affected by PM, and the regional proportional change is show in Fig. 4. For wheat and rice,  the proportional enhancement in crop production due to PM simulated with pDSSAT is very similar to that found in the offline analysis

10    (Fig. 4). This is true globally and within each region. The pDSSAT scenario with no stress is closely related to the offline analysis, which was unrestricted in production enhancement, so this good comparison is expected.

Unlike for wheat and rice, the proportional increase in maize production due to PM simulated by the pDSSAT model

[revised manuscript text omitted]

enhances the increase in crop production. pDSSAT could be configured to run in series over numerous years, as done by Liu et al. (2010a), to simulate the long term impacts on nitrogen cycling, but the uncertainty regarding the timing and retention of nitrogen deposited onto soils would remain, especially if not evaluated against observations.

5    If water stress is removed, the proportional enhancement of nitrogen deposition on crop production is slightly higher, as shown in Fig. 7. The largest change is for wheat, which is more water stressed than maize and rice in the model globally.

**5 Discussion and Conclusions**

10    To our knowledge, this is the first effort to integrate atmospheric air quality inputs into the dynamic simulation of a crop model. While ozone and PM air pollution have been incorporated into models which examine plant productivity, a crop model is needed to quantify the impacts on crop yield (not total biomass), the critical factor for understanding food security. This study takes into account crop-specific effects using the individual characteristics and distribution of each crop and the air pollution specific to the timeframe when each crop is grown. In this way, we produce a better constrained assessment of the impacts of

15 PM (radiation) and nitrogen deposition on crop production.

   Using restrictions on water and nitrogen availability and physiological limitations from the crop model provides a more realistic estimate of the impact of PM on crop production than in our earlier work which considered no such restrictions (Schiferl and Heald, 2018) . Maize production increases by only 2.3 % due to PM (11.5 % in Schiferl and Heald

20 (2018) ) using the max $\Delta$RUE = 100 % relationship, while wheat increases by 11.0 % (16.4 %) and rice increases by 3.4 % (8.9 %). The positive effect of PM on crop production is lessened when considering realistic restrictions to crop growth, but remains significant throughout the globe, especially in northern China. While it is difficult to compare across studies with varying approaches and metrics, our results are consistent in sign with the change in the $CO_2$ sink for crops due to PM found by Niyogi et al. (2004) and the global GPP change due to PM found by Strada and Unger (2016), noting that

25 $CO_2$ and GPP are not necessarily consistent with crop yield. We also find similar enhancements on a regional scale as Strada and Unger (2016), but for different regions, China and India in our case, as we do not consider the forested areas which dominate their results. For maize and wheat, the proportional increase in production is larger than the NPP increase found for all vegetation in China by Yue and Unger (2017). Our crop model is generally less responsive to PM than those enhancements found in forests in the locations studied by Cirino et al. (2014) and Strada et al. (2015), which is consistent with the smaller

30 canopies of crops. However, we are inconsistent in sign with the negative response in GPP due to PM found by Strada et al. (2015), although their study convolves croplands with grasslands.

Given that PM is simulated using current emissions (2010), these enhancements are already folded into present-day crop production, and may therefore be important to consider for air quality policy decisions which would reduce PM and thereby reduce  production in areas with crops sensitive to PM. For example, the decline in PM associated with the recent decrease in US $SO_2$ emissions has been shown to reduce US GPP by over 1 % since 1995 (Keppel-Aleks and Washenfelder, 2016). While this amount is small and aggregated for productivity over a large area, the impact of future PM change may be larger and more important to consider over a concentrated, highly polluted area. However, we note that our  results assume the maximum sensitivity of crops to PM and therefore the impact of PM on food production may be more modest, especially when considering secondary effects of PM (e.g., hydrological, meteorological) which may offset such enhancements.  More laboratory work is needed  to understand how different crop varietals respond to changes in radiation throughout the growing season.

 Our coupling of an atmospheric chemistry model with a crop model also provides an opportunity to explore the impact of atmospheric nitrogen deposition on crop production. We find that the impact of nitrogen deposition on crop production is significant, but more modest than the effect of PM. Our results are consistent with Thornton et al. (2007), who find ~2 % enhancement of global GPP due to nitrogen deposition. For crop yield, the impact of nitrogen deposition we find is also consistent in sign with Liu et al. (2013) over China. We underpredict the effect of nitrogen deposition on crops compared to the metric of sourced nitrogen content used by Ladha et al. (2016). This may be due to our relatively short assumed nitrogen deposition time frame. The fate of nitrogen in soil in managed ecosystems is a key uncertainty in estimating the response of crop production to changing atmospheric nitrogen deposition.

A  future with enhanced fertilizer inputs to feed growing populations will, if applied in excess, increase nitrogen inputs through deposition as well, potentially enhancing crop production further . At the same time, lower future $NO_x$ emissions are likely due to regulatory efforts, which will reduce the nitrogen deposition flux. These reductions could also reduce PM in areas prone to ammonium nitrate formation. The future trajectory of nitrogen deposition and PM remain uncertain, and thus the net impact on global crop production is unclear. An increased understanding of the implications of nitrogen deposition on crop production may also lead to better optimization of fertilizer application in areas where this impact is substantial.

The crop model responses to DF and nitrogen deposition examined in this study are uncertain and may vary from year to year. More work is needed, particularly controlled laboratory studies, to understand and evaluate these responses. It is critical to develop realistic crop models with reliable sensitivity to environmental factors to understand the pressure on future food security. Crop models tuned to reproduce observed yields without accounting for PM impacts (both direct and secondary) and nitrogen deposition may be less reliable under future levels of air pollution.

**Data availability**

The GC-RT and pDSSAT model data used in this study are archived at MIT and are available on request from the authors (schiferl@mit.edu). Emissions inventories implemented in GEOS-Chem v10-01 are available at https://github.com/GCST/hemco_data_download (GEOS-Chem Support Team, 2015). The DSSAT and pSIMS models and input data are available through http://www.dssat.net and http://www.github.com/RDCEP/psims, respectively (Elliott et al., 2014; Hoogenboom et al., 2015). The FAO GAEZ crop database is available at http://gaez.fao.org (FAO, 2016).

**Competing interests**

The authors declare that they have no conflict of interest.

**Acknowledgments**

Funding for this research was provided by the Martin Family Fellowship for Sustainability and the Abdul Latif Jameel World Water and Food Security Lab (J-WAFS) at the Massachusetts Institute of Technology (MIT). The authors thank the GEOS-Chem support staff and community for model documentation.

[Figure]

**Figure 1. (top row)** Crop production from base pDSSAT scenario (GC-RT SW only, no PM) with no stress applied for growing season ending in 2010. Difference in crop production due to **(second row)** water stress, **(third row)** nitrogen (N) stress, and **(bottom row)** both water and nitrogen stresses. For each row: **(left column)** maize, **(middle column)** wheat, and **(right column)** rice. Filtered for GAEZ base crop production greater than 0.01 Mg km$^{-2}$. Global production (top) or relative production change (second row-bottom) shown in upper right.

[Figure]

**Figure 2. Mean change in daytime (SW > 0) (top row) downward SW radiation and (bottom row) DF of the SW radiation at the surface due to PM from GC-RT. For pDSSAT growing season (determined by the base simulation) ending in 2010 for (left column) maize, (middle column) wheat, and (right column) rice. Filtered for GAEZ base crop production greater than 0.01 Mg km$^{-2}$.**

[Figure]

**Figure 3. Change in pDSSAT crop production due to PM with max $\Delta$RUE = 100 % with (top row) no stress and (bottom row) water and nitrogen (N) stresses applied. For growing season ending in 2010 for (left column) maize, (middle column) wheat, and (right column) rice. Filtered for GAEZ base crop production greater than 0.01 Mg km$^{-2}$. Global relative production change shown in upper right.**

[Figure]

**Figure 4. Regional relative change in crop production due to PM with max ∆RUE = 100 %: offline analysis from Schiferl and Heald (2017) (blue bars with hatching), pDSSAT simulation with no stress (dark blue bars), and pDSSAT simulation with water and nitrogen stresses (light blue bars). Change due to nitrogen (N) deposition in orange. For growing season ending in 2010 for (top row) maize, (middle row) wheat, and (bottom row) rice. Regions with a base production lower than 5 % of the global total are not shown.**

[Figure]

**Figure 5.** For (top row) pDSSAT base production and (bottom row) production with increased maximum kernels per plant under no stress: (left column) maize production with no PM, (middle column) maize production with PM (with max $\Delta$RUE = 100 %), and (right column) maize production due to PM. Filtered for GAEZ base crop production greater than 0.01 Mg km$^{-2}$. Global production (left and middle columns) or relative production change (right column) shown in upper right.

[Figure]

**Figure 6.** (top row) Total nitrogen (N) deposition from GEOS-Chem and (bottom row) reduced nitrogen (NH$_x$) fraction of this total. For pDSSAT growing season ending in 2010 (determined by the base simulation) for (left column) maize, (middle column) wheat, and (right column) rice. Filtered for GAEZ base crop production greater than 0.01 Mg km$^{-2}$.

[Figure]

**Figure 7. Change in pDSSAT crop production due to nitrogen (N) deposition with (top row) Nnitrogen stress and (bottom row) water and Nnitrogen stresses applied. For growing season ending in 2010 for (left column) maize, (middle column) wheat, and (right column) rice. Filtered for GAEZ base crop production greater than 0.01 Mg km⁻². Global relative production change shown in upper right.**

---

## Author Comment (AC2) · 31 May 2018

Response to Anonymous Referee #1

Note: page and line references mentioned in author changes refer to positions within the revised manuscript below.

General Comments: The goals of this paper are to answer questions about the impact of PM and N deposition on crop productivity, and specifically, how water, nitrogen, and plant physiological limitations contribute to the impact that PM and N deposition have on crops. These questions are important to address within the scientific community in order to increase our understanding of how anthropogenic activities affect food production. However, the introduction needs to better describe the current state of knowledge about PM and N deposition impacts on plant productively. To make the paper's arguments compelling, and to provide the reader with enough background knowledge about the state of research on PM and N deposition impacts on plant growth, the authors need to include a much more substantial and accurate discussion of current literature. In addition, more details about the simulations and how the model handles the N cycle could be added to help the reader interpret the impact of different variables on crop production. Similarly, it would be helpful to add a discussion section or place the manuscript's results in context to other literature, including other modeling and field studies.

**We thank the reviewer for these constructive comments. As detailed below, we have expanded our background discussion, added more details about the simulations, and included more context for our results with reference to the literature.**

Specific Comments: 1. Some of the claims in the introduction are too strong, and seem to contradict the current state of research. For example, on Page 2, lines 2-3, authors say that research has been done on the impact of climate and air quality on climate without considering physiological limitations and effects of water and nutrients. However, many disciplines examine how and why changes in crop production occur with climate change or changes in resource availability, particularly with regards to water use efficiency and nitrogen use efficiency.

**We agree that work has been done on the impacts of climate change on crops including some of these issues, but there has been limited work on the physiological limits to air quality impacts on crop growth. We have modified the introduction to more clearly specify the gaps in research.**

2. Most of the papers explaining the impacts of diffuse light on plant production are missing from the introduction, including those that discuss specific impacts on crops. For example, papers by Dev Niyogi (doi: 10.1029/2004GL020915), Kaicun Wang ( doi: 10.1029/2008GL034167, and Xiaoliang Lu (doi: 10.1016/j.agrformet.2017.02.002). Work by Gretchen Keppel-Aleks (doi: 10.1002/2016GL070052) may also be useful to discuss in this manuscript.

**The discussion of the impacts of diffuse light on plants and crops has been significantly expanded on page 2, line 18-page 3, line 19.**

3. Much more literature explaining the role of N deposition on plant and crop production also needs to be included in the introduction. There is a long history of research about the different responses plants have to N deposition and why this may occur.

**More literature about nitrogen cycling and nitrogen deposition has been added on page 3, line 27-page 4, line 23.**

4. What kinds of physiological limitations/caps do the authors have in mind when they discuss this in the introduction (e.g., Rubisco/chlorophyll production, root growth, cellular physiology)?

**The physiological limitations we have in mind include larger scale processes, such as the rate and magnitude of carbon allocation to plant carbon pools (roots, leaves, stems, grain) and life cycle timing. Cellular physiology is not represented in the crop model. This has been added on page 2, line 5.**

5. Can the authors provide more details about how crop growth is calculated in pDSSAT? Or is the crop model only a RUE model?

**Our pDSSAT simulations of maize, wheat and rice use the CERES modules for crop growth. These were developed separately for maize, wheat, and rice, and they simulate the carbon and nitrogen pools, among other parameters, associated with the various plant parts (e.g., leaves, stems, roots, grain) throughout the growth stages of each crop type. Potential dry matter (carbon) production is determined as a function of the solar radiation, SW (see Eq. 1). The actual dry matter production at each time step is limited by the effects of non-optimal temperature, water stress, and/or nitrogen stress, if applicable. Water and nitrogen stresses are determined by comparing the requirements of each crop with the amount of each resource available to the plant. Dry matter produced is then distributed into the plant parts based on those associated with the growth stage at that time. The sensitivity of growth rates and physical limitations for each plant part during each growth stage is determined by the physiology of that crop and cultivar. We have added this description on page 6, line 22-page 7, line 2.**

6. What size PM is used in the model?

**GC-RT uses a log-normal size distributed bulk scheme for each PM type, the physical and optical properties of which are described in Heald et al. (2014) and are accounted for in the radiative transfer scheme. PM sizes in GC-RT span several orders of magnitude, with mean diameters that range, for example, from black carbon, 0.04 µm, to sulfate, 0.14 µm, to dust, 8 µm. We have added this description on page 5, lines 28-31.**

7. How was the land cover and proportion of rice, wheat, and maize chosen for the crop model?

**The yield from the crop model is scaled to production using the GAEZ area for each crop in each crop model grid box. This is mentioned on page 8, lines 24-26.**

8. Page 4, Line 31: Can the authors provide more details about how the data are regridded to match the crop model resolution?

**We use an area-weighted regridding scheme to match the GC-RT data with the crop model grid and resolution. This has been added on page 7, line 23.**

9. Page 5, Line 6: Why was PM not chosen to change evaporation, when diffuse light can also reduce air and leaf temperatures?

**We chose to isolate only the potential carbon production relationship for comparison with our previous work in Schiferl and Heald (2018). This statement has been added on page 8, lines 7-9.**

10. If the manuscript is discussing changes in global crop production, why do the maps in the fitures only include part of the world?

**We focus on the significantly productive areas of maize, wheat, and rice for clarity of the figures. Global numbers are also presented. This is described on page 8, lines 28-30 and is consistent with our earlier work (Schiferl and Heald, 2018).**

11. Page 7, Line 26: If PM includes nitrate and ammonium, how do the authors remove the confounding effect of PM on the effect of N deposition on crop productivity?

**Here we mean that no PM impacts on radiation are considered. Nitrogen deposited as PM is considered and accounted for. This has been corrected.**

12. Figure 2: Is the mean change in daytime SW and DF a daily mean or a mean
(or total reduction) in the 2010 growing season? How do the authors define growing season when different regions of the world have different growing season lengths?

**The mean change in daytime SW and DF is the mean of daily changes in SW and DF due to PM throughout the growing season. A range of planting dates is prescribed by the pDSSAT model as mentioned on page 7, line 11. The potential length of the growing season is determined by the life cycle characteristics of the particular crop. Actual planting and harvest dates in a given simulation vary based on the meteorological (e.g., GDD, timing of rainfall) and resource (e.g., fertilizer amount, irrigated v. rainfed) inputs. A portion of this description has been added on page 8, lines 21-24.**

**For Fig. 2 and Fig. 6, we use the growing season determined by the base simulation (no PM, no stresses, no nitrogen deposition). This has been added on page 9, line 31 and in the Fig. 2 and Fig. 6 caption.**

13. Page 7, Line 2: By offline analysis, are the authors referring to the base simulation?
If not, can they provide more details to the difference?

**The offline analysis referred to here is the result from Schiferl and Heald (2018), which did not use a crop model and where no resource or physiological restrictions were placed on the enhancement due to PM. We are comparing the base simulation in this study (using a crop model), which has no resource restrictions, to the result from the previous study (without a crop model). In the previous study, unlimited growth enhancement (or loss) is allowed and is determined by the accumulated PM impacts throughout the growing season. We have added clarifications to the text on page 10, lines 4-5.**

14. Page 7, Line 26: What was the amount of extra N added? Figure 6 shows the total amount, but it is not clear what the base case's N deposition was. How were these amounts chosen? Also, how were fertilizer amounts chosen for simulations and were these the same across the globe? Does this make it more difficult to separate out the effect of N deposition alone?

**There is no nitrogen deposition in the base simulation, because prior to our modifications pDSSAT did not account for nitrogen deposition, only fertilizer application. All nitrogen deposition comes from GC-RT. We now note this on page 11, lines 6-7. Geographically distributed fertilizer comes from You et al. (2012), as mentioned on page 7, line 12-15. We clarified their source here. Given these textual additions, it should now be clear that we do separate out the effect of nitrogen deposition alone as fertilizer is applied in all cases, and nitrogen deposition is applied in addition to this to test the effect of nitrogen deposition.**

15. Page 8, Lines 15-18: Why do the authors decide to use a fertilizer application of 30 days prior to planting?

**The timing of the fertilizer application is determined by the pDSSAT base simulation. The timing of nitrogen deposition (which we apply in the same method as fertilizer) is uncertain in the literature, as discussed on page 8, lines 13-15. We have added acknowledgment that 30 days is an arbitrary timeframe on page 8, line 15. We assess the sensitivity of this selection with a description on page 11, line 31-page 12, line 1. We now mention a possible method for better constraining this timeframe on page 12, lines 1-3.**

16. Page 8, Line 25: This claim that this manuscript is the first to integrate atmospheric air quality inputs into a dynamic crop model seems bold. Can the authors more specifically describe what kind of work modelers have previously done to provide context for how their own manuscript is unique? For example, there has been work done on ozone pollution on plant productivity.

**Yes, ozone and PM pollution have been incorporated into models which examine plant productivity. However, plant productivity does not relate directly to the impact on crop yield, the critical factor for understanding food security. Our study takes into account crop specific effects using the individual characteristics and distribution of each crop and the pollution specific to the timeframe when each crop is grown to produce a better constrained assessment of the impacts of PM (light) and nitrogen deposition on crop production. This is added on page 12, lines 11-15 and is supported by additional introductory information suggested above.**

17. It may also help strengthen the paper and help place the methods and results within the greater biogeochemistry discipline to include a discussion section with information about past research on the effect of N deposition and PM on modeled and/or field based forest or crop productivity.

**We have renamed Sect. 5 to be "Discussion and Conclusions", which now includes comparison of our results to other studies.**

Response to Anonymous Referee #2

Note: page and line references mentioned in author changes refer to positions within the revised manuscript below.

This paper represents an innovative advance in the interdisciplinary research area of air pollution impacts on global food production through linking the global aerosol model (GEOS-Chem) to a crop modeling framework (pDSSAT). The study is "part 2" follow-up of a recent paper accepted for publication: Schiferl, L. D. and Heald, C. L, Particulate matter air pollution offsets ozone damage to global crop production, Atmos. Chem. Phys. Discuss., 2018. AgMIP has not traditionally assessed air pollution impacts on crops so this study is a welcome addition to the literature. The paper is brilliantly written and all figures and graphs are suitable for Biogeosciences.

1. There is already extensive integrated research on air pollution impacts on crops and plants. For example, flux-based risk maps e.g. Mills et al., Global Change Biology, 2011. Much of this line of work already integrates physiological constraints holistically.

**Yes, the impacts of ozone air pollution on crop production has been extensively studied, such as that done in Mills et al. (2011). This is not true for the impacts of PM air pollution. The impacts and metrics used for investigating ozone air pollution are discussed and applied in Schiferl and Heald (2018). We have clarified this in the text on page 2, lines 2-4.**

2. It is difficult to understand how a crop model that does not consider water stress can be useful, or even exist in the published literature? Why was pDSSAT chosen for the study? There are process-based crop models available e.g. GLAM. Moreover, international process-based vegetation models, for instance CLM and JULES, now include crop functional types that will inherently account for water limitation. Perhaps I have misunderstood something here. Crops are very sensitive to water availability.

**The pDSSAT model does include water stress, but this can be "turned off", which is what we do in our base simulation, by design, in order to compare it to the unrestricted PM enhancement in our previous work. We subsequently examine the impact of including water stress, which we agree, is more realistic.**

3. Following from point (2), it is concerning that the first paper did not include any limitation on the diffuse radiation fertilization of crops. For example, any process based model that incorporates Farquhar-Ball-Berry photosynthesis-stomatal conductance equations will automatically include limitations on the diffuse radiation fertilization e.g. Yue and Unger, Aerosol optical depth thresholds as a tool to assess diffuse radiation fertilization of the land carbon uptake in China, ACP, 2017.

**We agree that this could be a concern, however our results (see below) do not suggest that this substantially impacts our results. There is a limitation in the daily enhancement (max dRUE) in our previous work, but no limitation over the entire growing season. Our accounting was meant to be simple to provide a computational baseline against which to**

compare more complex calculations such as those done in pDSSAT, where carbon is tracked throughout the plant growth stages. Given the consistency in proportional change in production due to PM between our base simulation in this study and in the previous study (see Fig. 4, left two columns in each region for wheat and rice), the simple approach in our previous work may not be that unrealistic (when assuming no resource restrictions). While papers such as Yue and Unger (2017) do use models that account for a limit on the diffuse radiation fertilization effect, they do not allow for simulation of the crop-specific impacts of PM on crop production such as those suggested in our study. Yue and Unger (2017) is now mentioned on page 2, lines 26-27.**

4. The aerosol and crop models have very different horizontal resolution. The aerosol model is fairly coarse (2degx2.5deg). How is the aerosol radiation output downscaled for the high resolution crop model?

**We use an area-weighted regridding scheme to match the GC-RT data with the crop model grid and resolution. This has been added on page 7, line 23. We do not apply any downscaling of the radiation or nitrogen deposition from GC-RT. The same value is applied across all crop model grid boxes which overlap with a GC-RT grid box.**

5. How does pDSSAT calculate growth, biomass and yield from the potential carbon?

**Our pDSSAT simulations of maize, wheat and rice use the CERES modules for crop growth. These were developed separately for maize, wheat, and rice, and they simulate the carbon and nitrogen pools, among other parameters, associated with the various plant parts (e.g., leaves, stems, roots, grain) throughout the growth stages of each crop type. Potential dry matter (carbon) production is determined as a function of the solar radiation, SW (see Eq. 1). The actual dry matter production at each time step is limited by the effects of non-optimal temperature, water stress, and/or nitrogen stress, if applicable. Water and nitrogen stresses are determined by comparing the requirements of each crop with the amount of each resource available to the plant. Dry matter produced is then distributed into the plant parts based on those associated with the growth stage at that time. The sensitivity of growth rates and physical limitations for each plant part during each growth stage is determined by the physiology of that crop and cultivar. We have added this description on page 6, line 21-page 7, line 2.**

6. It appears that only the direct radiation changes of aerosols are calculated in GC_RT? How do aerosol indirect effects (aerosol-cloud interactions) influence the results and conclusions? Aerosol indirect radiative effects are widely accepted to be larger than the direct effects.

**Yes, the impacts of PM described in this study account only for the direct radiation changes through light absorbing and scattering effects, rather than secondary feedbacks associated with clouds or reduced radiation reaching the surface (e.g., hydrology, temperature). The relationships between the direct radiation changes and light use efficiency we do use are themselves uncertain. By design, we do not confound the results**

**and uncertainty further by assessing the indirect effects. A statement to this effect has been added on page 6, lines 4-7.**

7. The paper is lacking in any observational evidence and as such is a theoretical modeling study. Other studies have attempted to present observational evidence e.g. Strada and Unger, Observed aerosol-induced radiative effect on plant productivity in the eastern United States, Atmos Env 2015 and references therein; Strada and Unger, Potential sensitivity of photosynthesis and isoprene emission to direct radiative effects of atmospheric aerosol pollution, ACP, 2016; Yue et al., Future inhibition of ecosystem productivity by increasing wildfire pollution over boreal North America, ACP, 2017. One basic question is: do agricultural workers report that crops grow better and yield is higher under hazy and cloudy conditions?

**We have added references to the suggested studies which develop relationships with observations, usually of AOD and GPP, rather than the radiation values themselves and their impacts on crop production used in our study. We could not readily find reports from agricultural workers about crops growing under hazy and cloud conditions. Diffusivity screens, however, are applied to the windows of greenhouses to enhance production in such horticultural environments.**

8. Aerosol radiation changes will impact meteorology and hydrology (esp. temperature and soil moisture). These impacts likely have a much larger effect on crop productivity than the light changes e.g.: Yue et al., Ozone and haze pollution weakens net primary productivity in China, ACP, 2017. Is it possible that the light change impacts are overestimated in the current approach? For instance, an 11% enhancement in global wheat would have major impacts on global economics.

**Our results must be interpreted in the context they are presented; we are not claiming that the presence of aerosols has led to an 11% enhancement in global wheat production, only that the impact of aerosols on direct/diffuse radiation has led to an 11% enhancement in global wheat production. Clearly these changes are occurring coincident with other indirect effects of aerosols, as well as other environmental and climatic changes. Our study provides an estimate of the sensitivity of crop growth to two specific environmental factors. Deconvolving the environmental influence on global crop growth trends would be a far more complex endeavor. Our results indicate not that an enhancement in production may occur, but that an 11% enhancement in wheat production due directly to light change may already exist and is supported by sufficient water and nitrogen resources. These points have been clarified on page 13, lines 1-3.**

**A main implication is that crop models tuned to reproduce observed yields without accounting for the PM impacts (both direct and secondary) and nitrogen deposition may be less reliable under future levels of air pollution. This has been added on page 13, lines 32-33.**

**Resource and physiological constraints on global crop production enhancements from atmospheric particulate matter and nitrogen deposition**

Luke D. Schiferl[1], Colette L. Heald[1,2], and David Kelly[3]

[1]Department of Civil and Environmental Engineering, Massachusetts Institute of Technology, Cambridge, Massachusetts, USA

[2]Department of Earth, Atmospheric and Planetary Sciences, Massachusetts Institute of Technology, Cambridge, Massachusetts, USA

[3]University of Chicago Computation Institute, Chicago, IL, USA

*Correspondence to*: Luke D. Schiferl (schiferl@mit.edu)

**Abstract.** Changing atmospheric composition, induced primarily by industrialization and climate change, can impact plant health and may have implications for global food security. Atmospheric particulate matter (PM) can enhance crop production through the redistribution of light from sunlight to shaded leaves. Nitrogen transported through the atmosphere can also increase crop production when deposited onto cropland by reducing nutrient limitations in these areas. We employ a crop model (pDSSAT), coupled to input from an atmospheric chemistry model (GEOS-Chem), to estimate the impact of PM and nitrogen deposition on crop production. In particular, the crop model considers the resource and physiological restrictions to enhancements in growth from these atmospheric inputs. We find that the global enhancement in crop production due to PM in 2010 under the most realistic scenario is 2.3 %, 11.0 %, and 3.4 % for maize, wheat, and rice, respectively. These crop enhancements are smaller than those previously found when resource restrictions were not accounted for. Using the same model setup, we assess the effect of nitrogen deposition on crops and find modest increases (~2 % in global production for all three crops). This study highlights the need for better observations of the impacts of PM on crop growth and the cycling of nitrogen throughout the plant-soil system to reduce uncertainty in these interactions.

**1 Introduction**

Population growth is intensifying stress on global food production. Simultaneously, anthropogenic activities are changing many aspects of the earth system. This reinforces the need to better understand how crop production may be affected by changes to the water, air, light, and soil required for efficient growth. For example, Challinor et al. (2014) suggest a global decline in crop yield due to climate change of more than 10 % is likely by 2050.  This is uncertain, however, and the projected sign and magnitude varies by crop and region due to localized changes in factors such as temperature and precipitation combined with global carbon dioxide ($CO_2$) enhancement (IPCC, 2014). Many studies have explored the impacts of climate and air quality on crop production, both globally and regionally, with various results

depending on the tools, methods and processes used by each (e.g., Burney and Ramanathan, 2014; Lobell and Burke, 2010; Shindell et al., 2011; Tai et al., 2014). Investigations of the impacts of air quality on crops, in particular, have focused mainly on the negative impact of ozone pollution (Avnery et al., 2011; Mills et al., 2011; Van Dingenen et al., 2009). In comparison, only limited work has been conducted to assess how atmospheric particulate matter (PM) impacts crop production, and this

5  has  been done without considering physiological limitations (e.g., rate and magnitude of carbon pool allocation) and other environmental stresses (e.g., water and nutrients) (Greenwald et al., 2006; Schiferl and Heald, 2018).

Emitted through combustion and natural processes and formed through chemical

10  oxidation in the atmosphere, PM is the leading cause of air quality issues globally and is responsible for over 4 million premature deaths per year (Cohen et al., 2017). PM also impacts crop production by modifying shortwave radiation reaching the surface. Through the scattering of light, PM decreases the total shortwave (SW) radiation at the Earth's surface, which is made up of direct and diffuse light (SW = direct + diffuse). PM also increases the diffuse fraction (DF) of this SW radiation $(DF = \frac{diffuse}{SW})$. Increased DF more evenly distributes light throughout the canopy of a plant, redirecting light away from (at

15  times over-saturated) leaves in direct sunlight and onto shaded leaves. In this way, plants can more efficiently make use of incoming solar radiation (Kanniah et al., 2012).

Forests dominate previous studies of the impact of PM on plant productivity. Using network observations, Niyogi et al. (2004) showed that the $CO_2$ sink, a measure of plant productivity, increases with PM, indicated by aerosol optical depth (AOD), over

20  forests, but decreases for grasslands. More recent work related satellite AOD measurements to observations from $CO_2$ flux towers to quantify the impact of diffuse light on plant productivity. In the Amazon forest, enhancements in net ecosystem exchange (NEE) of up to 29 % are observed when the DF reaches approximately 0.5 (Cirino et al., 2014). Strada et al. (2015) find an increase in midday gross primary productivity (GPP) of ~13 % in US deciduous forests when DF is 0.4–0.6. Advanced canopy or leaf-scale process modelling has been used to further examine how PM impacts natural vegetation and the carbon

25  cycle. The model framework of Strada and Unger (2016) shows little sensitivity in global total GPP (~1–2 %) to PM pollution, with regional enhancements of ~5–8 % in North America and Eurasia and ~2 % in the Amazon, where forested canopies dominate. In China, Yue and Unger (2017) use AOD thresholds along with satellite observations and vegetation modelling to find the impact of PM pollution on net primary production (NPP) varies spatially from –3 % to +6 %. When accounting for the direct impacts of PM on light, temperatures, and hydrology, Yue et al. (2017) find a net increase in NPP of 5 %.

The impact of PM on managed vegetation (crops) is less well studied than for natural vegetation. PM can increase growth and production of crops when the increase in efficiency outweighs the loss of SW radiation. This depends on the local light conditions (changes in SW vs. DF) and crop type ($C_3$ vs. $C_4$). $C_4$ crops such as maize are less likely to be light

saturated than $C_3$ crops such as wheat. Niyogi et al. (2004) find that the $CO_2$ sink increases over croplands. In contrast with their forest sites, Strada et al. (2015) find a decrease in midday GPP of ~17 % associated with high observed AOD for a combination of US croplands and grasslands sites. They attribute this difference to canopy architecture which minimizes leaf shading, and thus the impact of diffuse light, when the sun is overhead. Greenwald et al. (2006) use relationships between DF, determined by climatological AOD, and a crop's radiative use efficiency (RUE), a measure of how effective a plant converts light into carbon, from Sinclair et al. (1992) along with varying meteorology and a crop model to estimate the impact of PM on crop yield. Assuming no restrictions on growth due to stresses at several locationssites, they find a large variation in impacts based on the DF-to-ΔRUE relationship chosen. Under the maximum relationship, maize increases by 0–10 %, wheat increases by 0–5 % and rice increases by 0–40 % under varying cloud conditions (Greenwald et al., 2006). Using this approach, but with a combined atmospheric chemistry and radiative transfer model to better simulation spatial and temporal variability of PM impacts on radiation, Schiferl and Heald (2018)Schiferl and Heald (2017) estimate a global positive impact of PM of 12 %, 16 %, and 9 % on maize, wheat, and rice production, respectively, for the year 2010. While their study uses a simple representation of the PM impacts on crop productivity, their approach isolates the impact of PM on crop production, which is not easily estimated based on previous observational analyses or mechanistic models. Observed AOD impacts on radiation are convolved with the influence of clouds, and as it is difficult to isolate only PM impacts, we cannot easily translate the observed AOD-to-carbon flux relationships to the impacts of DF on RUE. Mechanistic model studies account for all land biomass, but while crops do have smaller canopies and are less sensitive to DF then trees, such models do not differentiate between individual crop characteristics (e.g., canopies, growing seasons). Furthermore, the observed and simulated changes in NEE and GPP in these studies do not correspond directly to crop production, but rather on carbon uptake or release.

Industrial agriculture, driven by the need to produce food for a growing human population, has modified the global nitrogen (N) cycle (e.g., Bouwman et al., 2013; Smil, 1999). By artificially fixing inert nitrogen gas into reactive forms, humans have increased the fluxes of nitrogen throughout the environment, including into the atmosphere, onto land, and into the water (Galloway and Cowling, 2002). Nitrogen species in the atmosphere, both reduced and oxidized, return to the surface through deposition processes after being transported away from source regions. Anthropogenic influences on this deposition ese fluxes change the nitrogen balance in land and water ecosystems. In natural systems, this can cause acidification and eutrophication, which negatively impacts the biosphere (Beem et al., 2010; Erisman et al., 2007). Nitrogen accumulation into ecosystems from deposition is the main driver leading to an overall reduction in biodiversity, and while other factors such as direct toxicity, soil acidification, and increased susceptibility to stress are secondary, they can be dominant locally (Bobbink et al., 2010). While remaining substantially higher than during preindustrial time, current rates of nitrogen deposition have recently declined over the US and Europe but are expected to increase in developing countries in the future (Lamarque et al., 2013). This will contribute to projected (for 2050) nitrogen surpluses in Africa and Latin America corresponding with increases in crop and livestock production (Bouwman et al., 2013).

By including the coupling of carbon and nitrogen in a land-surface model, Thornton et al. (2007) show that GPP is limited by the supply of nitrogen to the biosphere and simulated over 40 % less GPP than a case that does not include this limitation. This carbon-nitrogen coupling dampens the response of vegetation to $CO_2$ concentration increases by over 70 %. The addition of atmospheric nitrogen deposition in the coupled system increases global GPP by ~2 %. When integrated into a fully-coupled

5    earth system model, there is a decrease in carbon uptake from $CO_2$ fertilization and an increase in carbon uptake from climate warming from the interactions between carbon and nitrogen. This increase in carbon uptake is due to enhanced nitrogen mineralization in the soil from a higher rate of decomposition (Thornton et al., 2009). Thomas et al. (2013) show that these simulated carbon-nitrogen responses for forests are smaller than those observed. Their model modifications result in greater retention of nitrogen deposition in biomass and a tighter coupling between nitrogen deposition and rising atmospheric $CO_2$

10    concentration. The model better represents observations by increasing the aboveground carbon storage response to nitrogen deposition.

The deposited nitrogenNitrogen deposition can also impact crop production, by providing additional fertilization, increasing yields in areas which are nitrogen limited (Goulding et al., 1998). These areas include portions of Africa, South America,

15    India, and Eastern Europe (Liu et al., 2010b; Mueller et al., 2012). Liu et al. (2013) show that in China nitrogen deposition leads to increased nitrogen uptake in non-fertilized croplands, resulting in a small increase in yield (1 t ha$^{-1}$) derived from a nitrogen uptake to yield ratio. Lassaletta et al. (2014) develop relationships between observed total nitrogen input and crop yield on a countrywide basis, but they do not disaggregate the impacts of deposition saying only that the input from deposition is small, but not negligible. While Ladha et al. (2016) estimate that 613 % of nitrogenN contained in global maize, wheat, and

20    rice comes from deposited nitrogen, to date, there has been no global study of the change of yield associated with nitrogen deposition, with most studies concentrating on the impacts of nitrogen deposition on interactions with atmospheric $CO_2$ and carbon storage. Folberth et al. (2016) neglect nitrogen deposition in their study of soil and meteorological data uncertainties in crop models due to the lack of available deposition data in a form suitable for global crop models.

25    Finally, PM and nitrogen deposition are also connected: The release of excess nitrogen from fertilizer application and livestock production in the form of ammonia ($NH_3$) contributes to PM formation in the atmosphere under acidic conditions (Seinfeld and Pandis, 2006). Nitric acid ($HNO_3$), an oxidized form of nitrogen oxides ($NO_x$) emissions from mobile and industrial sources, contributes both to the nitrogen burden and these acidic conditions. Nitrogen can also be incorporated in PM as organic nitrates when biogenic volatile organic compounds (BVOCs) react with $NO_x$ (Mao et al., 2013). Recent global modeling

30    studies incorporate more complex nitrogen transformations and cycling, such as the implementation of bi-directional ammonia fluxes into atmospheric chemistry models (Zhu et al., 2015) and climate-dependant agricultural nitrogen pathways into earth system models (Riddick et al., 2016).

Schiferl and Heald (2018)Schiferl and Heald (2017) quantify the impact that air quality (ozone and PM) has on current and future global crop production. Theiris analysis, while consistent with the approach generally applied to estimate air quality impacts on crops in previous studies mentioned above (e.g., Shindell et al., 2011; Tai et al., 2014; Van Dingenen et al., 2009), fails to account for the set of physical and biological restrictions placed on crop growth and production. In particular, they consider crop production enhancement due to the diffuse effect of PM is considered to be unlimited. However, water and nitrogen stresses and physiological caps placed on crop production may dampen these responses. In tThis study is a direct follow-up to Schiferl and Heald (2018), where we employ a crop production model to simulate the enhancements in crop production associated with PM and nitrogen deposition simulated by an atmospheric chemistry and radiative transfer model and explore the potential impact of resource and physiological constraints on this production.

**2 GEOS-Chem Atmospheric Chemistry Model**

The GEOS-Chem model (www.geos-chem.org, last access: June 2015) simulates the global concentration of gases and particles in three dimensions. Simulated PM concentrations are read into the Rapid Radiative Transfer Model for GCMs (RRTMG) to estimate the impact of PM on radiation throughout the atmosphere (Heald et al., 2014). Together these models are referred to as GC-RT. The model version and setup used here is the same as for the standard 2010 emissions scenario described by Schiferl and Heald (2018)Schiferl and Heald (2017). In brief: GC-RT is run at $2° \times 2.5°$ horizontal resolution using GEOS-5 meteorology for the years 2009 and 2010 from the NASA Global Modeling and Assimilation Office (GMAO). In this study, PM refers to the sum of all simulated aerosol species: sulfate ($SO_4^{2-}$), nitrate ($NO_3^-$), ammonium ($NH_4^+$), black carbon (BC), organic carbon (OC), sea salt and dust. Inorganic aerosol thermodynamics are coupled to an ozone–VOC–$NO_x$–oxidant chemical mechanism, where ISORROPIA II (Fountoukis and Nenes, 2007) handles the gas-particle phase partitioning of ammonium nitrate. GC-RT simulates wet and dry deposition of both aerosols and gases (Amos et al., 2012; Liu et al., 2001; Wang et al., 1998; Zhang et al., 2001). Major global anthropogenic gas emissions come from the Emission Database for Global Atmospheric Research (EDGAR) v4.2 ($NO_x$, carbon monoxide (CO), sulfur dioxide ($SO_2$)) (edgar.jrc.ec.europa.eu), the Reanalysis of the TROpospheric chemical composition (RETRO) inventory (non-methane VOCs) (Hu et al., 2015), and the Global Emission Inventory Activity (GEIA) inventory ($NH_3$). These are overlaid by regional inventories where available (see Schiferl and Heald (2018)Schiferl and Heald (2017) for details). Additional $NO_x$ emissions are from lightning and soil, described by Murray et al. (2012) and Hudman et al. (2012), respectively. Directly emitted aerosol sources include anthropogenic black carbon (BC) and organic carbon (OC) (Bond et al., 2007; Leibensperger et al., 2012), dust (Fairlie et al., 2007), and sea salt (Jaeglé et al., 2011). GC-RT uses a bulk aerosol scheme, where each aerosol species is described by a fixed log-normal size distribution, the physical and optical properties of which are described in Heald et al. (2014) and are accounted for in the radiative transfer scheme. PM sizes in GC-RT span several orders of magnitude, with mean diameters that range, for example, from black carbon, 0.04 µm, to sulfate, 0.14 µm, to dust, 8 µm.

In this study, we use hourly output of surface SW radiation and the diffuse and direct portions of this SW radiation from GC-RT both with and without PM under all-sky (real time variation in cloudiness) conditions. These are used to calculate the DF of the SW radiation.  While SW and DF respond differently to the differing properties of each PM

5   type, here we consider the net effect of all PM. The impacts of PM described in this study account only for the direct radiation changes through light absorption and scattering, and do not consider secondary feedbacks of aerosol on clouds, meteorology, and hydrology. We also use daily output of nitrogen deposition flux from the atmosphere, including the wet and dry deposition simulated for all nitrogen species. Nitrogen mass deposited from five species, ammonia, ammonium, nitric acid, nitrate, and nitrogen dioxide ($NO_2$), make up 98 % of the total simulated nitrogen deposition for 2010. Both the PM impacts on surface

10  radiation and the nitrogen deposition flux from the atmosphere are derived from the same GC-RT simulation, providing consistency over the emissions, chemistry, and deposition schemes described above.

**3 pDSSAT Crop Model**

**3.1 Model Description**

We use  the Decision Support System for Agrotechnology Transfer (DSSAT) v4.6 crop system model (Hoogenboom

15  et al., 2015), along with  the parallel System for Integrating Impact Models and Sectors (pSIMS) v2.0 (Elliott et al., 2014), together called pDSSAT, to simulate the global production of maize, wheat, and rice. DSSAT provides a unified interface which combines various crop simulation models (Jones et al., 2003). Inherently a point model, DSSAT uses daily meteorological data (minimum temperature, maximum temperature, precipitation, solar radiation, wind speed, and relative humidity) along with soil and management information at a given location. The model

20  then calculates a crop yield at harvest taking into account soil-plant-atmosphere dynamics throughout the growing season. Plant growth, in our case, is determined by the Crop-Environment Resource Synthesis (CERES) model module for each crop . CERES modules, developed separately for maize, wheat, and rice, simulate the carbon and nitrogen pools, among other parameters, associated with the various plant parts (e.g., leaves, stems, roots, grain) throughout the growth stages of each crop type. Potential dry matter (carbon) production is determined as

25  a function of the solar radiation, SW (see Eq. 1). The actual dry matter production at each time step is limited by the effects of non-optimal temperature, water stress, and/or nitrogen stress, if applicable. Water and nitrogen stresses are determined by comparing the requirements of each crop with the amount of each resource available to the plant. Dry matter produced is then distributed into the plant parts based on those associated with the growth stage at that time. The sensitivity of growth rates and physical limitations for each plant part during each growth stage is determined by the physiology of that crop and cultivar

30  (Jones et al., 1986; Ritchie et al., 1998; Ritchie and Otter, 1985). The simulation of these individual plant parts, rather than only total carbon, is of critical importance for this study as we are concerned with the production of grain to address impacts on food security. A recent review of CERES performances for maize, wheat, and rice finds that the models reproduce observed

grain yield well, with relative errors of ~10%, ~20%, and ~10%, respectively (Basso et al., 2016). They also find that secondary parameters such as soil temperature and nitrogen cycling were much less well represented.

[revised manuscript text omitted]

Nitrogen stress occurs when the plant tissue nitrogen concentration is less than the critical nitrogen concentration determined to provide optimal growth. In our base simulation, nitrogen stress follows different patterns compared to water stress for many regions and crops, although the global magnitudes in production reduction are similar. This response to carbon-nitrogen coupling is similar in sign and magnitude as that found for global GPP by Thornton et al. (2007). Maize production is affected by nitrogen stress primarily in the US Plains and Midwest. Nitrogen stress for wheat is distributed into all regions, while the effect on rice production is again lowest globally, it is and largest in southeast (SE) Asia. We note that the apparent impact of nitrogen stress on maize in Midwestern US is magnified by the large crop area in this region. Nitrogen stress is more similar from year to year in the model as fertilizer application, which provides nitrogen to the soil, and inherent soil nitrogen content is identical for all simulation years. Small variations do exist as variable temperatures and radiation impact the onset of crop growth stages and use of nitrogen. Folberth et al. (2016) find that uncertainty in soil data can impact simulated crop yield variability more than meteorological variability, especially for no water stress (irrigated), high nitrogen stress areas. In contrast, they find that irrigated areas with high nitrogen inputs show little difference between yield due to soil and meteorological input variability. Total production change due to both water and nitrogen stress does not combine linearly. This illustrates the interconnected system simulated by the crop model. Overall, these environmental and management constraints greatly reduce global crop production from its unstressed potential. They are important to consider when analyzing the impact of PM and nitrogen deposition on crop production.

**4 Results**

**4.1 Impact of Particulate Matter on Crop Growth**

To simulate the effect of PM on crop production, we run pDSSAT as above (for 2009 and 2010, sampling to the growing season ending 2010) with SW and DF input from GC-RT with and without PM. The differences in SW and DF due to PM over the pDSSAT growing season (determined by the base simulation) are shown in Fig. 2. PM has a negative effect on SW everywhere and positive effect on DF. The largest influence of PM is over China for all three crops. The influence is especially

noticeable for wheat, where a growing season over the winter corresponds with higher PM concentrations. The difference between the simulations with and without PM is the change in production due to PM, and this is shown in Fig. 3. We perform this procedure first with no stress factors applied in order to compare to the results found in Schiferl and Heald (2018), referred to here as the "offline analysis". The offline analysis uses a relativistic methodology which allows for unlimited growth

5    enhancement (or loss) and is determined by the accumulated PM impacts throughout the growing season.  In this pDSSAT simulation with no stress applied, global maize production increases by 1.7 %, wheat increases by 17.0 %, and rice increases by 6.2 %. Wheat production in the India and China+SE Asia regions is most affected by PM, and the regional proportional change is show in Fig. 4. For wheat and rice,  the proportional enhancement in crop production due to PM simulated with pDSSAT is very similar to that found in the offline analysis

10     (Fig. 4). This is true globally and within each region. The pDSSAT scenario with no stress is closely related to the offline analysis, which was unrestricted in production enhancement, so this good comparison is expected.

Unlike for wheat and rice, the proportional increase in maize production due to PM simulated by the pDSSAT model

[revised manuscript text omitted]

enhances the increase in crop production. pDSSAT could be configured to run in series over numerous years, as done by Liu et al. (2010a), to simulate the long term impacts on nitrogen cycling, but the uncertainty regarding the timing and retention of nitrogen deposited onto soils would remain, especially if not evaluated against observations.

5    If water stress is removed, the proportional enhancement of nitrogen deposition on crop production is slightly higher, as shown in Fig. 7. The largest change is for wheat, which is more water stressed than maize and rice in the model globally.

**5 Discussion and Conclusions**

10   To our knowledge, this is the first effort to integrate atmospheric air quality inputs into the dynamic simulation of a crop model. While ozone and PM air pollution have been incorporated into models which examine plant productivity, a crop model is needed to quantify the impacts on crop yield (not total biomass), the critical factor for understanding food security. This study takes into account crop-specific effects using the individual characteristics and distribution of each crop and the air pollution specific to the timeframe when each crop is grown. In this way, we produce a better constrained assessment of the impacts of
15   PM (radiation) and nitrogen deposition on crop production.

Using restrictions on water and nitrogen availability and physiological limitations from the crop model provides a more realistic estimate of the impact of PM on crop production than in our earlier work which considered no such restrictions (Schiferl and Heald, 2018) . Maize production increases by only 2.3 % due to PM (11.5 % in Schiferl and Heald
20   (2018) ) using the max $\Delta$RUE = 100 % relationship, while wheat increases by 11.0 % (16.4 %) and rice increases by 3.4 % (8.9 %). The positive effect of PM on crop production is lessened when considering realistic restrictions to crop growth, but remains significant throughout the globe, especially in northern China. While it is difficult to compare across studies with varying approaches and metrics, our results are consistent in sign with the change in the CO$_2$ sink for crops due to PM found by Niyogi et al. (2004) and the global GPP change due to PM found by Strada and Unger (2016), noting that
25   CO$_2$ and GPP are not necessarily consistent with crop yield. We also find similar enhancements on a regional scale as Strada and Unger (2016), but for different regions, China and India in our case, as we do not consider the forested areas which dominate their results. For maize and wheat, the proportional increase in production is larger than the NPP increase found for all vegetation in China by Yue and Unger (2017). Our crop model is generally less responsive to PM than those enhancements found in forests in the locations studied by Cirino et al. (2014) and Strada et al. (2015), which is consistent with the smaller
30   canopies of crops. However, we are inconsistent in sign with the negative response in GPP due to PM found by Strada et al. (2015), although their study convolves croplands with grasslands.

Given that PM is simulated using current emissions (2010), these enhancements are already folded into present-day crop production and may therefore be important to consider for air quality policy decisions which would reduce PM and thereby reduce production in areas with crops sensitive to PM. For example, the decline in PM associated with the recent decrease in US SO$_2$ emissions has been shown to reduce US GPP by over 1 % since 1995 (Keppel-Aleks and Washenfelder, 2016). While this amount is small and aggregated for productivity over a large area, the impact of future PM change may be larger and more important to consider over a concentrated, highly polluted area. However, we note that our results assume the maximum sensitivity of crops to PM and therefore the impact of PM on food production may be more modest, especially when considering secondary effects of PM (e.g., hydrological, meteorological) which may offset such enhancements. More laboratory work is needed to understand how different crop varietals respond to changes in radiation throughout the growing season.

Our coupling of an atmospheric chemistry model with a crop model also provides an opportunity to explore the impact of atmospheric nitrogen deposition on crop production. We find that the impact of nitrogen deposition on crop production is significant, but more modest than the effect of PM. Our results are consistent with Thornton et al. (2007), who find ~2 % enhancement of global GPP due to nitrogen deposition. For crop yield, the impact of nitrogen deposition we find is also consistent in sign with Liu et al. (2013) over China. We underpredict the effect of nitrogen deposition on crops compared to the metric of sourced nitrogen content used by Ladha et al. (2016). This may be due to our relatively short assumed nitrogen deposition time frame. The fate of nitrogen in soil in managed ecosystems is a key uncertainty in estimating the response of crop production to changing atmospheric nitrogen deposition.

A future with enhanced fertilizer inputs to feed growing populations will, if applied in excess, increase nitrogen inputs through deposition as well, potentially enhancing crop production further. At the same time, lower future NO$_x$ emissions are likely due to regulatory efforts, which will reduce the nitrogen deposition flux. These reductions could also reduce PM in areas prone to ammonium nitrate formation. The future trajectory of nitrogen deposition and PM remain uncertain, and thus the net impact on global crop production is unclear. An increased understanding of the implications of nitrogen deposition on crop production may also lead to better optimization of fertilizer application in areas where this impact is substantial.

The crop model responses to DF and nitrogen deposition examined in this study are uncertain and may vary from year to year. More work is needed, particularly controlled laboratory studies, to understand and evaluate these responses. It is critical to develop realistic crop models with reliable sensitivity to environmental factors to understand the pressure on future food security. Crop models tuned to reproduce observed yields without accounting for PM impacts (both direct and secondary) and nitrogen deposition may be less reliable under future levels of air pollution.

**Data availability**

The GC-RT and pDSSAT model data used in this study are archived at MIT and are available on request from the authors (schiferl@mit.edu). Emissions inventories implemented in GEOS-Chem v10-01 are available at https://github.com/GCST/hemco_data_download (GEOS-Chem Support Team, 2015). The DSSAT and pSIMS models and input data are available through http://www.dssat.net and http://www.github.com/RDCEP/psims, respectively (Elliott et al., 2014; Hoogenboom et al., 2015). The FAO GAEZ crop database is available at http://gaez.fao.org (FAO, 2016).

**Competing interests**

The authors declare that they have no conflict of interest.

**Acknowledgments**

Funding for this research was provided by the Martin Family Fellowship for Sustainability and the Abdul Latif Jameel World Water and Food Security Lab (J-WAFS) at the Massachusetts Institute of Technology (MIT). The authors thank the GEOS-Chem support staff and community for model documentation.

[Figure]

**Figure 1. (top row)** Crop production from base pDSSAT scenario (GC-RT SW only, no PM) with no stress applied for growing season ending in 2010. Difference in crop production due to **(second row)** water stress, **(third row)** nitrogen (N) stress, and **(bottom row)** both water and nitrogen stresses. For each row: **(left column)** maize, **(middle column)** wheat, and **(right column)** rice. Filtered for GAEZ base crop production greater than 0.01 Mg km⁻². Global production (top) or relative production change (second row-bottom) shown in upper right.

[Figure]

**Figure 2. Mean change in daytime (SW > 0) (top row) downward SW radiation and (bottom row) DF of the SW radiation at the surface due to PM from GC-RT. For pDSSAT growing season (determined by the base simulation) ending in 2010 for (left column) maize, (middle column) wheat, and (right column) rice. Filtered for GAEZ base crop production greater than 0.01 Mg km⁻².**

[Figure]

**Figure 3. Change in pDSSAT crop production due to PM with max ΔRUE = 100 % with (top row) no stress and (bottom row) water and nitrogen (N) stresses applied. For growing season ending in 2010 for (left column) maize, (middle column) wheat, and (right column) rice. Filtered for GAEZ base crop production greater than 0.01 Mg km⁻². Global relative production change shown in upper right.**

[Figure]

**Figure 4. Regional relative change in crop production due to PM with max ΔRUE = 100 %: offline analysis from  Schiferl and Heald (2018)** (blue bars with hatching), pDSSAT simulation with no stress (dark blue bars), and pDSSAT simulation with water and nitrogen stresses (light blue bars). Change due to nitrogen (N) deposition in orange. For growing season ending in 2010 for (top row) maize, (middle row) wheat, and (bottom row) rice. Regions with a base production lower than 5 % of the global total are not shown.

[Figure]

**Figure 5.** For (top row) pDSSAT base production and (bottom row) production with increased maximum kernels per plant under no stress: (left column) maize production with no PM, (middle column) maize production with PM (with max $\Delta$RUE = 100 %), and (right column) maize production due to PM. Filtered for GAEZ base crop production greater than 0.01 Mg km$^{-2}$. Global production (left and middle columns) or relative production change (right column) shown in upper right.

[Figure]

**Figure 6.** (top row) Total nitrogen (N) deposition from GEOS-Chem and (bottom row) reduced nitrogen (NH$_x$) fraction of this total. For pDSSAT growing season ending in 2010 (determined by the base simulation) for (left column) maize, (middle column) wheat, and (right column) rice. Filtered for GAEZ base crop production greater than 0.01 Mg km$^{-2}$.

[Figure]

**Figure 7. Change in pDSSAT crop production due to nitrogen (N) deposition with (top row) Nnitrogen stress and (bottom row) water and Nnitrogen stresses applied. For growing season ending in 2010 for (left column) maize, (middle column) wheat, and (right column) rice. Filtered for GAEZ base crop production greater than 0.01 Mg km⁻². Global relative production change shown in upper right.**